# Deep Kernel Relative Test for Machine-generated Text Detection

**Yiliao Song**[1*], **Zhenqiao Yuan**[2*], **Shuhai Zhang**[3], **Zhen Fang**[4], **Jun Yu**[5†], **Feng Liu** [2†*]

School of Computer and Mathematical Sciences, The University of Adelaide, Adelaide, AU[1]
School of Computing and Information Systems, University of Melbourne, Melbourne, AU[2]
School of Software Engineering, South China University of Technology, Guangzhou, CN[3]
Australian Artificial Intelligence Institute, University of Technology Sydney, Sydney, AU[4]
School of Intelligence Science and Engineering, Harbin Institute of Technology, Shenzhen, CN[5]
`{yiliao.song,zhenqiaoyuan02,shuhaizhangshz,fengliu.ml}@gmail.com`
`zhen.fang@uts.edu.au,yujun@hit.edu.cn`

## Abstract

Recent studies demonstrate that two-sample test can effectively detect machine-generated texts (MGTs) with excellent adaptation ability to texts generated by newer LLMs. However, two-sample test-based detection relies on the assumption that human-written texts (HWTs) must follow the distribution of seen HWTs. As a result, it tends to make mistakes in identifying HWTs that deviate from the *seen HWT* distribution, limiting their use in sensitive areas like academic integrity verification. To address this issue, we propose to employ *non-parametric kernel relative test* to detect MGTs by testing whether it is statistically significant that the distribution of *a text to be tested* is closer to the distribution of HWTs than to the MGTs' distribution. We further develop a *kernel optimisation* algorithm in relative test to select the best kernel that can enhance the testing capability for MGT detection. As relative test does not assume that a text to be tested must belong exclusively to either MGTs or HWTs, relative test can largely *reduce the false positive error* compared to two-sample test, offering significant advantages in practice. Extensive experiments demonstrate the superior performance of our method, compared to state-of-the-art non-parametric and parametric detectors. The code and demo are available: `https://github.com/xLearn-AU/R-Detect`.

## 1 Introduction

The advent of large language models (LLMs) such as GPT-3 (Brown et al., 2020) has demonstrated their remarkable performance in text generation across various applications, *e.g.*, text summarization (Liu & Lapata, 2019; Luo et al., 2023), dialogue generation (Li et al., 2016; Lancaster, 2023), and machine translation (Bahdanau et al., 2014; Lee, 2023). However, their misuse raises concerns, particularly regarding the generation of fake content (Zellers et al., 2019), plagiarism (Lee et al., 2023; Stokel-Walker, 2022), and other ethical issues (Weidinger et al., 2021). The increasingly indistinguishable machine-generated texts (MGTs) produced by newer LLMs aggravates worries about authenticity (Lin et al., 2022) and accountability (Susnjak & McIntosh, 2024). Recent research further highlights the versatility of LLMs in generating domain-specific content that can even deceive domain experts (Else, 2023), necessitating reliable MGT detection techniques.

Existing post-hoc detectors are generally classified into three types: metric-based methods (Mitchell et al., 2023; Soto et al., 2024; Hans et al., 2024), classifier-based methods (Hu et al., 2023; Tian et al., 2024), and test-based methods (Zhang et al., 2024). Since metric-based and classifier-based methods are parametric, their performance inevitably depends on specific types of MGTs, limiting their adaptability. In contrast, the non-parametric test, theoretically supported by the kernel *two-sample test* (2ST) (Gretton et al., 2012a; Liu et al., 2020; 2021; Gao et al., 2021), ignores specific generation mechanisms and focuses solely on the intrinsic differences between human-written texts

---

*Equal contribution. †Correspondence to: yujun@hit.edu.cn, fengliu.ml@gmail.com.

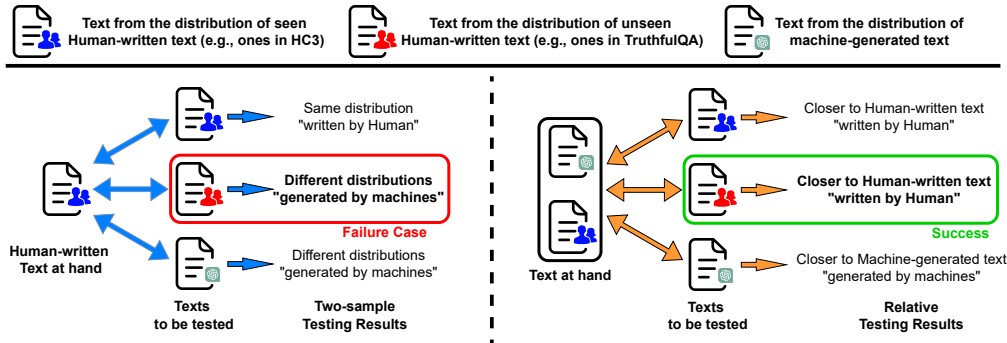

Figure 1: Difference between using two-sample testing (subfigure (a)) and relative testing (subfigure (b)) in detecting machine-generated texts. Blue arrows represent the two-sample testing procedures, and orange arrows represent the relative testing procedures.

(HWTs) and MGTs. Consequently, it performs well even on mixed texts generated by different LLMs or under varying LLM settings (*e.g.*, temperature, top-k sampling (Vilnis et al., 2023)).

However, using 2ST may result in a high false positive rate of marking HWTs as MGTs (Zhang et al., 2024). False identification of HWTs can lead to unnecessary content removal or unjust accusations, which is unacceptable in sensitive areas such as academic integrity verification (Dalalah & Dalalah, 2023). During detection, an MGT is identified by rejecting the null hypothesis that unauthored texts and HWT references are drawn from the same distribution (Zhang et al., 2024). Intuitively, the 2ST-based detection method assumes that the unauthored text must *exclusively belong* to either MGTs or HWTs—that is, the distributions of HWTs and MGTs should not have any overlap in principle. However, since in practical situations the distributions of HWTs and MGTs often overlap, the non-overlapping assumption may lead to a high false positive rate.

To address the challenge caused by non-overlapping assumption, we propose employing a *non-parametric kernel relative test* to determine whether a text is written by a machine or a human. Introduced by Bounliphone et al. (2016), the relative test can determine which of two samples is significantly more similar to a reference sample. Instead of using historical MGT or HWT samples as references, we use the unauthored text as the reference sample. By applying the relative test, we can determine whether the MGT sample or the HWT sample from the database is closer to the unauthored text, thereby making a detection decision. Since the relative test does not assume texts exclusively belong to either MGTs or HWTs, the false positive rate is significantly reduced compared to two-sample tests. This offers substantial advantages for practical applications.

*Selecting a suitable kernel* is crucial for non-parametric tests (Gretton et al., 2012a; Sutherland et al., 2017; Liu et al., 2020; 2021), especially when dealing with complex data (Liu et al., 2020; Zhang et al., 2024). However, the kernel selection issue is rarely explored in the context of relative tests. Here we propose a novel method to optimize kernels to make non-parametric kernel relative tests more powerful in determining whether a reference sample is closer to HWT or MGT. Specifically, by empirical studies, we discover that the kernels performing well in 2ST also consistently perform well in relative tests. Therefore, following Liu et al. (2020), we select the kernel for relative test by increasing the test power of 2ST. Experimental results show that the optimal kernel-based relative test significantly outperforms those based on common-used kernels (*i.e.*, Gaussian kernels).

**Motivation of this study.** This study introduces a non-parametric post-hoc method for detecting MGTs by framing the detection task as a relative test problem. Aiming for an *interpretable* and *fine-grained* MGT detection to ensure the ethical use of LLMs (Kumar et al., 2023), we consider whether *MGT is detectable in practice*, which has been affirmed by studies (Chakraborty et al., 2024; Hans et al., 2024). Consequently, we adopt the assumption from Zhang et al. (2024) that *MGTs and HWTs are distinguishable in distribution*. Although the 2ST-based MGT detection method (Zhang et al., 2024) offers statistical interpretability, it is limited by assuming texts belong exclusively to either MGTs or HWTs. Furthermore, the choice of using MGT or HWT as a reference can yield different detection results. These limitations *motivate* our use of relative tests for MGT detection in this study.

**Contribution of this study:**

- The potential of using statistical hypothesis tests for MGT detection is explored, enriching the detection framework with robust theoretical foundations derived from hypothesis testing.
- The MGT detection task is conceptualized as a relative test problem, providing enhanced detection accuracy and flexibility compared to the traditional two-sample test method.
- A novel method to optimize kernels in relative tests for MGT detection is proposed, significantly improving the effectiveness and efficiency of the detection process.
- Superior detection performance is demonstrated across various LLM settings, clearly outperforming state-of-the-art non-parametric and parametric MGT detectors.

## 2  PROBLEM SETUP AND RELATED NOTATIONS

### 2.1  PROBLEM SETUP

As reviewed in section 1, the question of whether *MGT is detectable in practice* has been evidenced by previous studies (Chakraborty et al., 2024; Hans et al., 2024). Based on this conclusion, we consider the case where MGTs and HWTs originate from two non-identical text spaces.

**Text Space.** Let $\mathcal{S}$ be the space of all possible texts. We consider HWTs to belong to the subspace $\mathcal{S}_{\mathrm{h}} \subset \mathcal{S}$ and MGTs belong to another subspace $\mathcal{S}_{\mathrm{m}} \subset \mathcal{S}$.

> **Problem 1** (MGT Detection). *MGT detection aims to find a detector $f : \mathcal{S}_{\mathrm{h}} \cup \mathcal{S}_{\mathrm{m}} \to \{MGT, HWT\}$, which effectively distinguishes between MGTs and HWTs.*

To find the $f$, we consider only assume that $\mathcal{S}_{\mathrm{h}} \cap \mathcal{S}_{\mathrm{m}} = \emptyset$ otherwise the same $X$ will have two different $Y$s. For example, supposing the word "cat" is written by a human while another "cat" is generated by a machine, $f$ cannot detect whether this "cat" is generated by a human or machine. To demonstrate the impact of this overlap ratio on our designed method, we conduct a toy study to analyse how the detection performance of our method varies by the overlap ratio in Appendix D.

In addition, we define $f$ as the composition of a feature transformation function $g$ and a feature detector $D$, that is, $f = D \circ g$. In practice, $g$ employs techniques such as pre-trained transformers (Liu, 2019) to convert the original text into textual representations. Consequently, the MGT-detection problem is transformed into a task of analyzing these textual representations, with the detector aiming to determine whether a text's representations originate from human-written texts.

To estimate the performance of MGT detector $f$, we consider the true positive rate (TPR) and false positive rate (FPR) as the metrics, *i.e.*,

$$\mathrm{TPR} = P_s \left[ f(s) = \mathrm{MGT} \mid s \in \mathcal{S}_m \right] \quad \uparrow$$
$$\mathrm{FPR} = P_s \left[ f(s) = \mathrm{MGT} \mid s \in \mathcal{S}_h \right] \quad \downarrow$$

We expect that the MGT detector can achieve a high TPR while maintaining a low FPR.

### 2.2  NOTATIONS AND CONCEPTS

**Maximum Mean Discrepancy.** Maximum mean discrepancy (MMD) (Gretton et al., 2012a) comparing samples from distribution, aims to measure the closeness between two distributions.

**Definition 1** (Maximum Mean Discrepancy (Gretton et al., 2012a)). *Let $k : \mathcal{X} \times \mathcal{X} \to \mathbb{R}$ be the bounded kernel, $\mathbb{P}$ and $\mathbb{Q}$ be Borel probability measures on $\mathcal{X} \subset \mathbb{R}^d$. Given $\mathbf{x}$ and $\mathbf{x}'$ independent observations from $\mathbb{P}$, and $\mathbf{y}$ and $\mathbf{y}'$ independent observations from $\mathbb{Q}$, the squared MMD is*

$$\mathrm{MMD}^2 (\mathbb{P}, \mathbb{Q}; k) = \mathbb{E}_{\mathbf{x}, \mathbf{x}'} \left[ k\left(\mathbf{x}, \mathbf{x}'\right) \right] + \mathbb{E}_{\mathbf{y}, \mathbf{y}'} \left[ k\left(\mathbf{y}, \mathbf{y}'\right) \right] - 2\mathbb{E}_{\mathbf{x}, \mathbf{y}} \left[ k(\mathbf{x}, \mathbf{y}) \right].$$

Here $\mathrm{MMD}^2$ refers to the population discrepancy. However, we can only observe the sample from distributions in practice. Following theorem shows that the unbiased empirical estimation of $\mathrm{MMD}^2 (\mathbb{P}, \mathbb{Q}; k)$ exists, meaning the population $\mathrm{MMD}^2 (\mathbb{P}, \mathbb{Q}; k)$ can be estimated by finite samples. This provide an effective way for us to estimate the population $\mathrm{MMD}^2$ via sample data.

**Theorem 1** (Gretton et al. (2012a)). *Define independent identically distributed (i.i.d) observations $X_m := \{\mathbf{x}_i\}_{i=1}^m \sim \mathbb{P}^m$ and $Y_m := \{\mathbf{y}_j\}_{j=1}^m \sim \mathbb{Q}^m$. Let $\mathcal{V} := \{\mathbf{v}_l\}_{l=1}^m$ be observations with $\mathbf{v}_i := (\mathbf{x}_i, \mathbf{y}_i)$. Then the unbiased empirical estimates of $\mathrm{MMD}^2 (\mathbb{P}, \mathbb{Q}; k)$ is:*

$$\mathrm{MMD}_u^2(X_m, Y_m; k) = \frac{1}{m(m-1)} \sum_{i \neq j}^m h\left(\mathbf{v}_i, \mathbf{v}_j\right), \tag{1}$$

where $h\left(\mathbf{v}_i, \mathbf{v}_j\right) = k(\mathbf{x}_i, \mathbf{x}_j) - k(\mathbf{x}_i, \mathbf{y}_j) - k(\mathbf{y}_i, \mathbf{x}_j) + k(\mathbf{y}_i, \mathbf{y}_j)$. *Given* $\mathbb{P} \neq \mathbb{Q}$ *and* $\mathbb{E}\left(h^2\right) < \infty$, $\mathrm{MMD}_u^2(X_m, Y_m; k)$ *converges in distribution to a Gaussian according to*

$$\sqrt{m}\left(\mathrm{MMD}_u^2(X_m, Y_m; k) - \mathrm{MMD}^2(\mathbb{P}, \mathbb{Q}; k)\right) \xrightarrow{\mathcal{D}} \mathcal{N}\left(0, \sigma_{XY}^2\right),$$

*where*

$$\sigma_{XY}^2 = 4\left(\mathbb{E}_\mathbf{v}\left[\left(\mathbb{E}_{\mathbf{v}'} h\left(\mathbf{v}, \mathbf{v}'\right)\right)^2\right] - \left[\mathbb{E}_{\mathbf{v}, \mathbf{v}'}\left(h\left(\mathbf{v}, \mathbf{v}'\right)\right)\right]^2\right). \tag{2}$$

**Relative Test.** Relative test (Bounliphone et al., 2016) comparing the similarity of two candidate distributions to a reference distribution, aims to determine which candidate is closer to the reference.

**Definition 2** (Relative Test). *Let* $\mathbf{x}$, $\mathbf{y}$, $\mathbb{P}$ *and* $\mathbb{Q}$ *be defined as above;* $\mathbf{z}$ *be an observation with distribution* $\mathbb{Z}$. *Given i.i.d observations* $X_m = \{\mathbf{x}_i\}_{i=1}^m \sim \mathbb{P}^m$, $Y_n = \{\mathbf{y}_j\}_{j=1}^n \sim \mathbb{Q}^n$ *and* $Z_r = \{\mathbf{z}_l\}_{l=1}^r \sim \mathbb{Z}^r$, *and* $\mathbb{P} \neq \mathbb{Z}$, $\mathbb{Q} \neq \mathbb{Z}$, *relative test is the statistical hypothesis test that:*

*Null hypothesis:* $\mathrm{MMD}(\mathbb{P}, \mathbb{Z}; k) \leq \mathrm{MMD}(\mathbb{Q}, \mathbb{Z}; k),$

*Alternative hypothesis:* $\mathrm{MMD}(\mathbb{P}, \mathbb{Z}; k) > \mathrm{MMD}(\mathbb{Q}, \mathbb{Z}; k),$

and the $p$-values for testing null hypothesis versus alternative hypothesis are

$$p \leq \Phi\left(-\frac{\mathrm{MMD}_u^2(X_m, Z_r; k) - \mathrm{MMD}_u^2(Y_n, Z_r; k)}{\sqrt{\sigma_{XZ}^2 + \sigma_{YZ}^2 - 2\sigma_{XZYZ}}}\right), \tag{3}$$

where $\Phi$ is the *cumulative distribution function* (CDF) of a standard normal distribution, and $\sigma$ the covariance. Here, $\sigma_{YZ}^2$ and $\sigma_{XZ}^2$ are the variances of $\mathrm{MMD}_u^2(Y_n, Z_r; k)$ and $\mathrm{MMD}_u^2(X_m, Z_r; k)$ respectively (refer to Equation 2). The empirical estimation $\sigma_{YZXZ}$ is presented in Appendix A.

## 3 METHODOLOGY

### 3.1 RELATIVE TEST AS A SOLUTION

In this section, we propose relative test to search the detector $f$ as the solution of Problem 1. Let

$$X_m = \{\mathbf{x}_i\}_{i=1}^m \sim \mathbb{P}^m \ i.i.d. \text{ and } Y_m = \{\mathbf{y}_j\}_{j=1}^m \sim \mathbb{Q}^m \ i.i.d.$$

be observations from $\mathcal{S}_\mathrm{h}$ and $\mathcal{S}_\mathrm{m}$, respectively. The distribution $\mathbb{P}$ encapsulates the statistical characteristics of the seen HWTs, while $\mathbb{Q}$ represents the distribution containing statistical characteristics of seen MGTs. The distribution $\mathbb{Z}$ corresponds to the texts being tested.

When $\mathbb{P} \neq \mathbb{Z}$ and $\mathbb{Q} \neq \mathbb{Z}$, given a significance level $\alpha$, the relative test for MGT detection will classify a given text $s \in \mathcal{S}$ as MGT if the null hypothesis of the relative test,

$$\mathrm{MMD}(\mathbb{P}, \mathbb{Z}; k) \leq \mathrm{MMD}(\mathbb{Q}, \mathbb{Z}; k), \tag{4}$$

is rejected. Specifically, let $f(s; p, \alpha) := \mathbf{1}_{\{p < \alpha\}}$, then the decision rule is defined as

$$f(s) = \begin{cases} \mathrm{MGT} & \text{if } f(s; p, \alpha) = 1, \\ \mathrm{HWT} & \text{if } f(s; p, \alpha) = 0, \end{cases} \tag{5}$$

where $\mathbf{1}$ denotes the indicator function, and $p$ is the $p$-value shown in Equation 3. We give the implementation of relative test MGT detection (R-Detect) in Algorithm 1. The input of Algorithm 1 is $X_m$, $Y_m$, $g$, $k$, $\alpha$. $g$ is a fixed function that converts texts to textual representations. In this study, we fix $g$ as OpenAI's RoBERTa-based GPT-2 detector model (Liu, 2019) with more details discussed in Appendix B. $k$ is a given kernel function, either learning from samples by Algorithm 3 or using pre-assigned (see Appendix B). $\alpha$ is the threshold used for rejecting the null hypothesis. In default setting, $\alpha = 0.05$ and we also present results for a different $\alpha$ in Appendix C. Given these input, we first calculate the $\mathrm{MMD}_u^2$ values to come out the $p$-value. By comparing $p$ with $\alpha$, we can get the detection result as the output of Algorithm 1.

**Test Power *v.s.* TPR *v.s.* FPR.** In hypothesis testing, the test power is defined as the probability of rejecting the null hypothesis when the alternative hypothesis is true (Zhang et al., 2024). In R-Detect, the null hypothesis is formulated as

$$\mathrm{MMD}(\mathbb{P}, \mathbb{Z}; k) \leq \mathrm{MMD}(\mathbb{Q}, \mathbb{Z}; k),$$

which implies that the text to be tested is an MGT. Consequently, the test power in R-Detect given that the ground truth is an MGT, is the probability that an MGT is correctly identified as an MGT, corresponding to TPR. Similarly, the test power in R-Detect given that the ground truth is a HWT, is the probability that an HWT is incorrectly identified as an MGT, corresponding to FPR. Specifically, we present the calculation of evaluation metrics for R-Detect in Algorithm 2, utilizing test power and the Area Under the Receiver Operating Characteristic curve (AUROC) (Jiménez-Valverde, 2012).

| **Algorithm 1** Relative Test MGT Detection | **Algorithm 2** R-Detect Evaluation |
|---|---|
| **Input:** $X_m, Y_m, g, k, \alpha$; | **Input:** $X_m, Y_m, g, k, \alpha$; |
| **Test Text:** $s \in \mathcal{S}$; | **Test Text:** $S_1 \subset \mathcal{S}_h, S_2 \subset \mathcal{S}_m$   % HWTs, MGTs |
| $r \leftarrow \mathrm{MMD}_u^2(g(X_m), g(s); k)$;   % Equation 1 | **for** $round = 1, 2, \ldots, n$ **do** |
| $h \leftarrow \mathrm{MMD}_u^2(g(Y_m), g(s); k)$;   % Equation 1 | Randomly Choose $\_S_h \subset S_1, \_S_m \subset S_2$ |
| $p \leftarrow \Phi(r, h)$;   % Equation 3 | $D_1, p_1 \leftarrow$ R-Detect($\_S_h$);   % Algorithm 1 |
| $f \leftarrow f(s; p, \alpha)$   % Equation 5 | $D_2, p_2 \leftarrow$ R-Detect($\_S_m$);   % Algorithm 1 |
| **if** $f == 1$ **then** | **end for** |
| **Output:** $s$ is machine-generated text | FPR$\leftarrow P_{d \in D_1} (d = 1)$; % test power given MGT |
| **else** | TPR$\leftarrow P_{d \in D_2} (d = 1)$; % test power given HWT |
| **Output:** $s$ is human-written text | AUROC$\leftarrow D_1, D_2, 1 - p_1, 1 - p_2$; |
| **end if**. | **Output:** TPR, FPR, AUROC |

## 3.2 KERNEL OPTIMISATION FOR RELATIVE TEST MGT DETECTION

In section 3.1, we provide a predefined kernel function as an input to Algorithm 1 for calculating the value of $\mathrm{MMD}_u^2$. This design allows R-Detect to generate detection results directly without requiring any training. Specifically, Bounliphone et al. (2016) employ a Gaussian kernel, where the bandwidth is determined by the median pairwise distance between data points. The *choice of kernel* can significantly impact the test power in non-parametric tests (Gretton et al., 2012b; Sutherland et al., 2017), especially when handling complex data types (Liu et al., 2020; 2021).

However, *how to select an optimal kernel* is rarely explored in relative test. Inspired by existing studies on kernel optimisation for kernel-based 2ST, we here derive our kernel optimisation for relative test from the empirical study that answers the following question:

"*Is it empirically feasible to empower relative test from a corresponding two sample test?*"

**MMD-based 2ST.** Kernerl-based method is a very popular class of non-parametric statistical tests (Berlinet & Thomas-Agnan, 2011). Using kernel-based MMD for the two-sample test has a history (Gretton et al., 2012b). Given $X_m = \{\mathbf{x}_i\}_{i=1}^m \sim \mathbb{P}^m$, $Y_n = \{\mathbf{y}_j\}_{j=1}^n \sim \mathbb{Q}^n$, MMD-based 2ST aims to determine whether $X_m$ and $Y_n$ are from the same distribution, that is, $\mathbb{P} = \mathbb{Q}$.

**Deep Kernel Optimisation.** The selection of kernel is always challenging as a good kernel can largely increase the test performance. Recent study proposes deep kernel, which builds a kernel with a deep network to enable optimising kernels function for MMD-based 2ST through maximizing its test power (Liu et al., 2020). We here consider the *deep kernel* technique (Liu et al., 2020):

$$k_\omega(\mathbf{x}, \mathbf{y}) = [(1 - \epsilon)\kappa(\phi_{\hat{f}}(\mathbf{x}), \phi_{\hat{f}}(\mathbf{y})) + \epsilon q(\hat{f}(\mathbf{x}), \hat{f}(\mathbf{y})), \tag{6}$$

where $\epsilon \in (0, 1)$, $\phi_{\hat{f}}(\mathbf{x}) = \phi(\hat{f}(\mathbf{x}))$ is a deep neural network with feature extractor $\hat{f}$, $\kappa$ and $q$ are Gaussian kernels with bandwidth $\sigma_\phi$ and bandwidth $\sigma_q$, respectively. Since $\hat{f}$ is fixed, the set of parameters of $k_\omega$ is $\omega = \{\epsilon, \phi, \sigma_\phi, \sigma_q\}$.

For the empirical use of 2ST in multiple population scenario (Zhang et al., 2024), we consider the *multi-population aware optimisation* for kernel-based MMD:

$$k_\omega^* = \arg \max_{k_\omega} \mathrm{MPP}(X_m, Y_m; k_\omega) / \sigma(X_m, Y_m; k_\omega), \tag{7}$$

where $\mathrm{MPP}(X_m, Y_m; \mathcal{H}_k) := \mathbb{E}[k_\omega(X_m, X_m') - 2k_\omega(X_m, Y_m)]$ and $\sigma(X_m, Y_m; k_\omega)$ is the squared root of variance for MMD, referring to Equation 2.

**Test Power for 2ST-based MGT Detection.** Let i.i.d. observations $R := (r_1, r_2, \ldots, r_m)$ be reference texts. MGT detection will mark a text $s$ as MGT in either case of *i*) given $R \subset \mathcal{S}_h$, the null hypothesis is rejected; or *ii*) given $R \subset \mathcal{S}_m$, the null hypothesis is not rejected. Similarly, it will mark $s$ as HWT in either case of *iii*) given $R \subset \mathcal{S}_m$, the null hypothesis is rejected; or *iv*) given $R \subset \mathcal{S}_h$, the null hypothesis is not rejected.

Table 1: Empirical analysis for test power of two-sample test in MGT detection.

| Test Power | $R^{\#\mathrm{HWT}}$ | $R^{\#\mathrm{MGT}}$ |
|---|---|---|
| $s^{\#\mathrm{MGT}}$ | TPR | FNR |
| $s^{\#\mathrm{HWT}}$ | FPR | TNR |

Correspondingly, the test power of the 2ST in MGT detection (*i.e.,* , the probability of rejecting $s$ and $R$ from the same distribution) are:

**i)**   Given $R \subset \mathcal{S}_h$, the rejection probability is TPR when $s$ is MGT;

**ii)**   Given $R \subset \mathcal{S}_h$, the rejection probability is FPR when $s$ is HWT;

**iii)**  Given $R \subset \mathcal{S}_m$, the rejection probability is FNR[1] when $s$ is MGT;

**iv)**  Given $R \subset \mathcal{S}_m$, the rejection probability is TNR[2] when $s$ is HWT.

The empirical analysis on test power of 2ST in MGT detection is presented in Table 1.

**Kernel Optimisation.** By comparing the empirical test power of R-Detect in section 3.1: Test Power v.s.TPR v.s. FPR with the 2ST case (Table 1), we conclude that the test power of the relative test is empirically equivalent to that of a two-sample test using HWTs as the reference for MGT detection. Therefore, we optimize a deep kernel that achieves the best test power in two-sample MGT detection and apply this kernel in R-Detect. Let $S_h^{tr}$ and $S_m^{tr}$ be collections of historical HWTs and MGTs used as the training set. Given $g$, the mapping function that converts text to textual representations, and $\lambda$, the hyperparameter for learning the deep kernel, the op-

---

**Algorithm 3** Kernel Optimisation in R-Detect

**Input:** $X \leftarrow X_m^{tr}, Y \leftarrow Y_m^{tr}, g, \lambda \leftarrow 10^{-8}$;
**Initialize:** $\omega$
**for** $t = 1, 2, \ldots, T$ **do**
 $k_\omega \leftarrow k_\omega(g(X), g(Y))$;  % Equation 6
 $M \leftarrow \mathbb{E}[k_\omega(X, X') - 2k_\omega(X, Y)]$;
 $s \leftarrow \sigma^2(\mathbb{P}, \mathbb{Q}; k_\omega)$;  % Equation 2
 $J_\omega \leftarrow M/\sqrt{s}$
 $\omega \leftarrow \omega + \lambda \nabla_{\text{Adam}} \hat{J}_\omega$;
**end for**
**Output:** $k_\omega$.  % Optimised kernel

---

timized kernel $k_\omega$ is learned in Algorithm 3. By inputting the $k_\omega$ learned from existing HWTs and MGTs into Algorithm 1 as $k$, we obtain the relative test MGT detection with the optimized kernel.

## 4 EXPERIMENTS

### 4.1 SETTTINGS

**Datasets and LLMs.** We design our experiments on data from five benchmarks: HC3 (Guo et al., 2023), TruthfulQA (TQA) (He et al., 2023; Lin et al., 2022), RAID (Dugan et al., 2024), and DetectRL (Wu et al., 2024). These benchmarks encompass various LLMs, covering both adversarial attacks and non-English scenarios. Further details are provided in Appendix B. We also manually generate MGTs by GPT-4o (OpenAI, 2024) from five randomly chosen human-written essays in the Essay dataset (Verma et al., 2024) with the prompt: "rewrite". Please refer to Appendix F for the texts we are using and its corresponding rewritten texts.

**Baselines.** We compare R-Detect to **five non-parametric baselines** and **seven parametric baselines**. The non-parametric baselines include two variations of our method 1) R-**Detect-**$k^m$: R-Detect with the Gaussian kernel optimised by median heuristic bandwidth (Bounliphone et al., 2016) and 2) R-**Detect w/o** $k^*$, namely our method without kernel optimisation; one state-of-the-art non-parametric MGT detector MPP with different settings (Zhang et al., 2024), *i.e.,* 3) **MPP-HWT**, 4) **MPP-MGT** and 5) **MPP-R**; seven state-of-the-art parametric detectors, *i.e.,* 6) **Detect-GPT** (Mitchell et al., 2023), 7) **Fast-DetectGPT** (Bao et al., 2024), 8) **DNA-GPT** (Yang et al., 2024a), 9) **Bino** (Hans et al., 2024) and its variation 10) **Bino-FPR**, 11) **DALD** (Zeng et al., 2024), and 12) **Text-Fluroscopy** (Yu et al., 2024). Appendix B.4 provides more details of each baseline.

**Evaluation Metrics.** We evaluate the detection performance using test power for 2ST (Gretton et al., 2012a) which is the TPR given MGT the ground-truth and false positive rate (FPR) given HWT the ground-truth. In addition, we evaluate the detection performance via AUROC (Jiménez-Valverde, 2012). In the default setting, we randomly take 512 tokens and repeat the experiments $10 \times 10$ times given a specific experimental design. We use **bold** numbers to indicate the best results in tables.

During evaluation, our method is implemented in a *zero-shot* setting unless otherwise noted. For *further implementation details*, please refer to Appendix B.

### 4.2 COMPARISON BETWEEN NON-PARAMETRIC MGT DETECTORS

#### 4.2.1 WHEN HWTS FROM SEEN HWT DISTRIBUTION

Here, we use the learned kernel function $k_\omega$ from HC3 to test unseen texts from HC3 to mimic the MGT-detection when HWTs are from the seen distribution of HWTs using the datasets of

---

[1] FNR $= P_s[f(s) = \text{HWT} \mid s \in \mathcal{S}_m]$
[2] FNR $= P_s[f(s) = \text{HWT} \mid s \in \mathcal{S}_h]$

Table 2: Test power ($p$) and AUROC on texts to be tested from HC3-MGT and HC3-HWT.

| Non-parametric Detectors | $tp^{\text{MGT}}$ | $tp^{\text{HWT}}$ | AUROC | Total time (s) |
|---|---|---|---|---|
| R-Detect-$k^m$ | $1.00_{\pm 0.00}$ | $0.50_{\pm 0.41}$ | $0.99_{\pm 0.02}$ | 285.35 |
| R-Detect w/o $k^*$ | $1.00_{\pm 0.00}$ | $0.82_{\pm 0.18}$ | $0.80_{\pm 0.37}$ | 192.66 |
| MPP-HWT | $1.00_{\pm 0.00}$ | $0.10_{\pm 0.12}$ | $0.72_{\pm 0.26}$ | **56.90** |
| MPP-MGT | $0.90_{\pm 0.15}$ | $0.00_{\pm 0.00}$ | $0.72_{\pm 0.24}$ | 62.59 |
| MPP-R | $1.00_{\pm 0.00}$ | $0.01_{\pm 0.03}$ | $0.99_{\pm 0.01}$ | 117.66 |
| R-Detect(Ours) | $\mathbf{1.00}_{\pm 0.00}$ | $\mathbf{0.00}_{\pm 0.00}$ | $\mathbf{1.00}_{\pm 0.00}$ | 90.40 |

HC3→HC3 in Appendix B. The test power $tp$ and AUROC are from Algorithm 2. In default experiment setting, the results of test power $tp$, AUROC and running time are presented in Table 2. Specifically, we separate the results of HC3-MGT and HC3-HWT for better understanding.

**Test Power on HC3-MGTs.** The test power on HC3-MGT $tp^{\text{MGT}}$ equals to the TPR in practice. A larger $tp^{\text{MGT}}$ denotes better performance. As is shown in Table 2, most of the non-parametric detectors achieve good performance with regards to $tp^{\text{MGT}}$.

**Test Power on HC3-HWTs.** the test power on HC3-HWT $tp^{\text{HWT}}$ equals to the TPR in practice. A smaller $tp^{\text{HWT}}$ denotes better performance. According to Table 2, R-Detect-$k^m$ and R-Detect w/o $k^*$ have large $tp^{\text{HWT}}$, means that they mistakenly label HWTs as MGTs. MPP-HWT is better than these two, but still has 10% probability of mislabeling HWTs. MPP-MGT has a $tp^{\text{HWT}}$ of 1% but it is unknown whether to choose HWT or MGT in the real case. Compared to them, MPP-R achieve 1% FPR and R-Detect archives 0% FPR constantly.

**AUROC on HC3.** For test-based detector, we used $1-p$-value as the prediction score for calculating AUROC because a smaller $p$-value means more likely to be labeled as MGT. Compared to the other baselines, R-Detect gets the best AUROC result of $\mathbf{1.00}_{\pm 0.00}$, which indicates that R-Detect can always assign a higher rejection probability for MGTs than HWTs. This has surpassed MPP-R by 1%↓ MPP-MGT or MPP-HWT by 28%↑.

**Running Time.** We have give more details of machine we used for running all the experiments in Appendix B. Compared to single-side MPP, MPP-HWT or MPP-MGT, R-Detect is slightly slower but faster than a naive relative MPP version. Overall, the detection is efficiency.

**Non-parametric Detectors with Varied Settings.** We vary the default setting by 1) changing the default $\alpha = 0.05$ to $\alpha = 0.90$; 2) changing the token size from 512 to 256; 3) limiting the length of reference data from all available HC3 expect for the test to the same length with token size. The comparison tables are in Tables 5–8. The result does not differ a lot. Therefore, the non-parametric methods are robust when HWT are from known HWT distribution.

### 4.2.2 WHEN HWTS FROM UNSEEN HWT DISTRIBUTION

Here, we use the learned kernel function from HC3 to test texts from TQA to mimic MGT-detection when HWTs are from unseen distributions. We test both cases of *against old LLM* and *against newer LLM* by using HC3→TQA-ChatGPT and HC3→TQA-GPT4 separately (Appendix B). The results are shown in Table 3. We also highlight how the result is different from the result in section 4.2.1-Table 2 which is for seen HWT distribution, using ↑ and ↓ to denote better or worse performance.

**Result of Detection against Old LLMs.** HC3 only contains MGT generated by ChatGPT. We found the detection performance does not decrease much when using kernel learned from HC3 to test TQA-MGT which is also generated from ChatGPT, only MPP-MGT slightly worse. However, the baselines have significantly worse performance for identifying TQA-HWT *i.e.,* human-written texts in TQA. In addition, comparing MPP-HWT with MPP-MGT, their false positive rate differs a lot with each other, where MPP-MGT can achieve a better FPR of $0.17$ but MPP-HWT is $0.88$. This indicates a big uncertainty of MPP's practical use as it is unknown whether MGT or HWT should be chosen for the reference in advance. Compared to the result for the case of seen HWT distribution, R-Detect still performs excellently, correctly marking all MGTs and HWTs from TQA.

**Result of Detection against Newer LLMs.** We also test the performance on GPT4-generated texts in TQA to test if the detection is valid when MGT is generated by a newer LLM. In the column HC3→TQA-GPT4, we can see the $tp^{\text{MGT}}$ does not drop much among baseline methods. This indicates the non-parametric method's adaptation capability to newer LLMs.

Table 3: Test power ($p$) and AUROC on texts to be tested from TQA-ChatGPT and TQA-GPT4.

| Non-parametric Detectors ($\uparrow\downarrow$ compared to Table 2) | HC3→TQA-ChatGPT | | | HC3→TQA-GPT4 | | |
|---|---|---|---|---|---|---|
| | $tp^{\text{MGT}}$ | $tp^{\text{HWT}}$ | AUROC | $tp^{\text{MGT}}$ | $tp^{\text{HWT}}$ | AUROC |
| R-Detect-$k^m$ | $1.00_{\pm 0.00}$ | $1.00_{\pm 0.00}\downarrow$ | $0.92_{\pm 0.05}\downarrow$ | $1.00_{\pm 0.00}$ | $0.76_{\pm 0.41}\downarrow$ | $0.83_{\pm 0.14}\downarrow$ |
| R-Detect w/o $k^*$ | $1.00_{\pm 0.00}$ | $0.98_{\pm 0.04}\downarrow$ | $0.90_{\pm 0.03}\uparrow$ | $1.00_{\pm 0.00}$ | $0.92_{\pm 0.18}\downarrow$ | $0.66_{\pm 0.06}\downarrow$ |
| MPP-HWT | $1.00_{\pm 0.00}$ | $0.88_{\pm 0.09}\downarrow$ | $0.77_{\pm 0.17}\uparrow$ | $0.76_{\pm 0.42}\downarrow$ | $0.87_{\pm 0.12}\downarrow$ | $0.69_{\pm 0.10}\downarrow$ |
| MPP-MGT | $0.814_{\pm 0.42}\downarrow$ | $0.17_{\pm 0.38}\downarrow$ | $0.73_{\pm 0.20}\uparrow$ | $0.92_{\pm 0.11}$ | $0.83_{\pm 0.00}\downarrow$ | $0.83_{\pm 0.09}\uparrow$ |
| MPP-R | $1.00_{\pm 0.00}$ | $0.96_{\pm 0.03}\downarrow$ | $0.52_{\pm 0.02}\downarrow$ | $1.00_{\pm 0.00}$ | $1.00_{\pm 0.03}\downarrow$ | $0.50_{\pm 0.00}\downarrow$ |
| R-Detect(Ours) | $\mathbf{1.00}_{\pm 0.00}$ | $\mathbf{0.00}_{\pm 0.00}$ | $\mathbf{1.00}_{\pm 0.00}$ | $\mathbf{1.00}_{\pm 0.00}$ | $\mathbf{0.00}_{\pm 0.00}$ | $\mathbf{1.00}_{\pm 0.00}$ |

Table 4: Comparison between Parametric and Non-parametric detection results.

| (Token Size= 512) | HC3→HC3 | | | HC3→TQA-ChatGPT | | | HC3→TQA-GPT4 | | | time (s) |
|---|---|---|---|---|---|---|---|---|---|---|
| | $tp^{\text{MGT}}$ | $tp^{\text{HWT}}$ | AUROC | $tp^{\text{MGT}}$ | $tp^{\text{HWT}}$ | AUROC | $tp^{\text{MGT}}$ | $tp^{\text{HWT}}$ | AUROC | |
| Bino | $1.00_{\pm 0.00}$ | $1.00_{\pm 0.00}$ | $1.00_{\pm 0.00}$ | $1.00_{\pm 0.00}$ | $1.00_{\pm 0.00}$ | $1.00_{\pm 0.00}$ | $1.00_{\pm 0.00}$ | $1.00_{\pm 0.00}$ | $1.00_{\pm 0.00}$ | 1064.36 |
| Bino-FPR | $1.00_{\pm 0.00}$ | $0.00_{\pm 0.00}$ | – | $1.00_{\pm 0.00}$ | $0.00_{\pm 0.00}$ | – | $1.00_{\pm 0.00}$ | $0.00_{\pm 0.00}$ | – | – |
| R-Detect(Ours) | $1.00_{\pm 0.00}$ | $0.00_{\pm 0.00}$ | $1.00_{\pm 0.00}$ | $1.00_{\pm 0.00}$ | $0.00_{\pm 0.00}$ | $1.00_{\pm 0.00}$ | $1.00_{\pm 0.00}$ | $0.00_{\pm 0.00}$ | $1.00_{\pm 0.00}$ | **180.25** |
| R-Detect$_{0.9}$ (Ours) | $1.00_{\pm 0.00}$ | $0.00_{\pm 0.00}$ | – | $1.00_{\pm 0.00}$ | $0.00_{\pm 0.00}$ | – | $1.00_{\pm 0.00}$ | $0.00_{\pm 0.00}$ | – | – |
| (Token Size= 256) | HC3→HC3 | | | HC3→TQA-ChatGPT | | | HC3→TQA-GPT4 | | | |
| Bino | $1.00_{\pm 0.00}$ | $1.00_{\pm 0.00}$ | $1.00_{\pm 0.00}$ | $1.00_{\pm 0.00}$ | $1.00_{\pm 0.00}$ | $1.00_{\pm 0.00}$ | $1.00_{\pm 0.00}$ | $1.00_{\pm 0.00}$ | $0.992_{\pm 0.016}$ | – |
| Bino-FPR | $1.00_{\pm 0.00}$ | $0.00_{\pm 0.00}$ | – | $1.00_{\pm 0.00}$ | $0.00_{\pm 0.00}$ | – | $0.00_{\pm 0.00}$ | $0.00_{\pm 0.00}$ | – | – |
| R-Detect(Ours) | $1.00_{\pm 0.00}$ | $0.00_{\pm 0.00}$ | $1.00_{\pm 0.00}$ | $1.00_{\pm 0.00}$ | $0.00_{\pm 0.00}$ | $1.00_{\pm 0.00}$ | $1.00_{\pm 0.00}$ | $0.00_{\pm 0.00}$ | $1.00_{\pm 0.00}$ | – |
| R-Detect$_{0.9}$ (Ours) | $1.00_{\pm 0.00}$ | $0.00_{\pm 0.00}$ | – | $1.00_{\pm 0.00}$ | $0.00_{\pm 0.00}$ | – | $1.00_{\pm 0.00}$ | $0.00_{\pm 0.00}$ | – | – |

**Summary.** The experimental results valid our claim that 1) recent non-parametric detection methods can adapt to different LLMs but 2) have increased false positive rate of mislabeling human-written text as MGT when it comes from unseen HWT distributions. Our proposed method, R-Detect, can maintain the good adaptation capability of 1) and address the limitation of 2) at the same time.

### 4.3 COMPARE TO PARAMETRIC METHODS: BINO AS AN EXAMPLE

In section 4.2, we compared R-Detect to a variety of non-parametric baselines. In this section, we present a comparison result between our method R-Detect and the state-of-the-art parametric MGT detector, Bino (Hans et al., 2024). In particular, we compare R-Detect and Bino with varied thresholds and token sizes. In R-Detect, the threshold is the $\alpha$ referring to the significance level for a statistical test while the threshold in Bino, is the classification threshold.

The results are represented in Table 4, As a highlight, the AUROC evaluation does not depend on threshold, and thus is the same among different thresholds. Therefore, we use "–" to denote this number is the same with R-Detectcan always perform very good even with a big variation of the threshold, from 0.05 to 0.9. However, compared to R-Detect, Bino's performance is very sensitive to the threshold selection. In addition, our method has surpassed Bino on AUROC by 0.8%$\uparrow$ on the HC3→TQA-GPT4 with a smaller token size. In addition, R-Detectshows a significant advantage with regards to the running time, taking less than 20% time of Bino's.

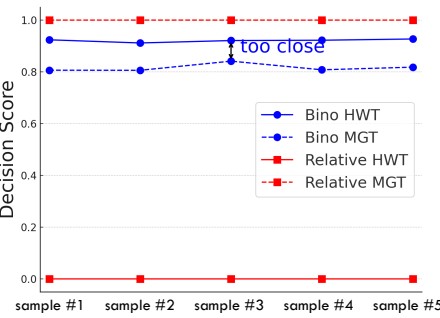

Figure 2: Decision score difference.

Specifically, we give the detection score for Bino and R-Detect in Figure 2. In R-Detect, we use the $1 - p$-value of the statistics calculated for relative test as the decision score. A smaller $p$-value denotes a larger probability of rejecting the null hypothesis. It can be seen, the $1 - p$-value is very close to 0 when the ground truth of the text is HWT, and very close to 1 when the ground truth of the text is MGT. This makes HWT and MGT distinguishable with a large range of threshold choices.

The score for bino is a metric based on contrasting two closely related language models. Bino will detect a text as MGT when the Bino score is less than a threshold. It can be seen that, Bino will have a good result if we choose the threshold between $(0.85, 0.9)$ but might induce ompletely wrong detection on MGT given the threshold larger than 0.85 or completely wrong detection on HWT given the threshold smaller than 0.8 — such as Bino's bad performance for HWT detection in Table 4.

### 4.4 CASE STUDY: DETECTING GPT4-REWRITTEN TEXTS

We here show a case study of using our method and Bino on the *rewritten texts by GPT4o*. We here only present our results against Bino's results (Hans et al., 2024), because MPP (Zhang et al., 2024) varies the decision when choosing HWT or MGT. As all the MGTs are rewritten texts by the newest GPT4-o model, it is very challenging to distinguish them from their human-written versions.

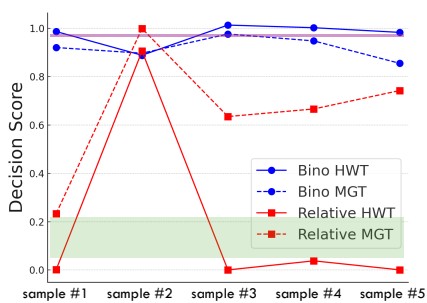

Figure 3: Decision score difference on GPT4-rewritten texts.

We list the decision score for R-Detect and Bino in Figure 3. It can be seen that although it is less distinct between HWT and rewritten-MGTs, it is feasible to find a threshold for R-Detect in the green shallow area to provide detection results with both acceptable TPR and FPR. However, it is less feasible for Bino to find such a threshold to get performance better than a random decision from existing knowledge (see the magenta shallow area, where we have tried the best given the ground truth).

### 4.5 PERFORMANCE EVALUATION ACROSS DIVERSE BASELINES AND BENCHMARKS

The analysis in section 4.2 demonstrates that, as both are testing-based methods, R-Detect offers a unique advantage over MPP (Zhang et al., 2024). Meanwhile, the analysis in section 4.3 aims to explain the benefits of non-parametric detectors compared to parametric ones. In this section, we extend our experimental analysis to more diverse and challenging settings: 1) MGTs are generated by multiple LLMs; 2) the text data may contain adversarial attacks; 3) the text data is written in a language other than English. The experimental results, analyzed in Appendix E, demonstrate that:

- R-Detect achieves the best performance in multi-LLM-generated texts (E.1);
- R-Detect is robust against various adversarial attacks (E.1);
- While our R-Detect is not always the best in non-English detection tasks in a zero-shot setting, detection effectiveness improves significantly with a properly optimised kernel (E.2).
- Using advanced feature extractors significantly improves R-Detect's performance (E.3).

## 5 RELATED WORKS

### 5.1 LLM-EMPOWERED MGTS: CONCERNS AND SOLUTIONS

Large language models such as ChatGPT (Schulman et al., 2022), Google's LaMDA (Thoppilan et al., 2022), Meta's OPT (Zhang et al., 2022), LLaMa (Touvron et al., 2023), and Falcon (Almazrouei et al., 2023), trained on enriched human text data, are capable of generating natural, fluent, and high-quality content. Their usage has surged dramatically due to easy public access (Watch, 2023); for example, since its launch in November 2022, monthly visits to ChatGPT have increased 15-fold (Singh, 2024). However, the increasing indistinguishability of LLM-generated texts from human-written content has raised growing concerns about their misuse (Weidinger et al., 2022), including phishing attacks (Hazell, 2023), disinformation (Zellers et al., 2019; Adelani et al., 2020), plagiarism (Lee et al., 2023; Stokel-Walker, 2022), and other ethical risks (Weidinger et al., 2021).

As humans can be easily deceived by MGTs (Ippolito et al., 2020; Zellers et al., 2019), developing effective MGT detectors is seen as a significant step toward ensuring the responsible use of generative language models (Dhaini et al., 2023). In terms of detection design prepared watermarking methods (Kirchenbauer et al., 2023; 2024; Yang et al., 2024b) aim to modify the distribution of generated text in a pre-designed manner, but this study does NOT focus on such methods. Instead, we focus on post-hoc detection, which assumes no interaction during the text generation process (Chakraborty et al., 2024). Serving as *tools rather than the ultimate goal*, finer-grained MGT detection is recommended to ensure the ethical use of LLMs (Kumar et al., 2023), posing new challenges for interpretable detection mechanism and explainable detection results.

## 5.2 PARAMETRIC POST-HOC MGT DETECTION

Recent studies introduce an amount of post-hoc detection methods, such as Fast-DetectGPT (Bao et al., 2024), DNA-GPT (Yang et al., 2024a), MPP (Zhang et al., 2024), MPU (Tian et al., 2024), Binoculars (Hans et al., 2024), and others (Soto et al., 2024). A parametric detection method assumes MGT is an output of a generative language model with specific parameters. Consequently, the detection performance is inherently tied to the particular type of MGT, limiting their adaptation capabilities across different generative language models. In general, current parametric methods can be divided into two categories: metric-based and classifier-based approaches.

**Metric-based MGT Detection.** These methods leverage pre-trained LLMs or scoring models to measure the statistical discrepancies between HWTs and MGTs. Commonly used metrics include log-likelihood (Solaiman et al., 2019), entropy (Ippolito et al., 2020), rank (Gehrmann et al., 2019), log-rank (Su et al., 2023), N-Gram (Yang et al., 2024a), and log probability (Mitchell et al., 2023; Bao et al., 2024; Hans et al., 2024). Since these metrics are often derived from pre-trained LLMs, they can facilitate zero-shot detection with proper design (Bao et al., 2024; Hans et al., 2024). However, these metric-based detection methods tend to suffer from inferior performance when there is a significant domain gap between the language of the generated text and the scoring model.

**Classifier-based MGT Detection.** Classifier-based methods typically involve training a classification model using both HWTs and MGTs (Mitrović et al., 2023). For example, OpenAI-D (Solaiman et al., 2019) fine-tunes a RoBERTa model on GPT-2-generated texts for detecting GPT-2 outputs. ChatGPT-D (Guo et al., 2023) employs two strategies (using either pure answered text or QA pairs) to train the model with the HC3 dataset. OpenAI has recently fine-tuned a GPT model (Kirchner et al., 2023) using data from Wikipedia, WebText, and human evaluations to develop a web interface for a discrimination task involving texts generated by 34 different language models.

## 5.3 NON-PARAMETRIC POST-HOC MGT DETECTION

**Two-sample Test-based MGT Detection.** A non-parametric approach, MPP is proposed to use two-sample test for MGT detection (Zhang et al., 2024). MPP optimises the kernel function in MMD (Gretton et al., 2012a; Liu et al., 2020) to determine whether the distribution of an unauthored text differs from that of a reference text. It assumes that HWTs and MGTs follow distinct distributions, regardless of how MGTs are generated, allowing it to adapt easily to MGTs generated by newer LLMs, *i.e.*, optimising kernels with ChatGPT-generated texts (OpenAI, 2022) while testing on GPT-Neo (Black et al., 2021), GPT-j-6b (Wang & Komatsuzaki, 2021), and GPT4all-j (Anand et al., 2023). Compared to parametric methods—whether metric-based (Solaiman et al., 2019; Gehrmann et al., 2019; Mitchell et al., 2023) or classifier-based (Solaiman et al., 2019; Guo et al., 2023)—MPP demonstrates superior detection performance (Zhang et al., 2024).

## 6 CONCLUSION

This paper presents a machine-generated text detector, R-Detect, that has been empirically proved, with extensive experiments, to have a small false alarm rate and a high successful rate in detecting texts generated by representative large language models (LLMs). Specifically, R-Detect is a non-parametric tool and does not require any knowledge regarding how the texts are generated by LLMs (i.e., we do not involve any LLM-based inference), which is suitable for many scenarios where only the information regarding texts is available. In addition, we also empirically compare R-Detect with existing detection tools in the literature, and the results show that R-Detect achieve state-of-the-art performance in terms of both of false alarm rate and detection rate. Notably, R-Detect can even outperform parametric detection tools that need to access LLMs or surrogate LLMs, which further justifies the significance of R-Detect, especially when using it in real-world applications.

### ACKNOWLEDGMENTS

ZQY and FL are supported by the Australian Research Council (ARC) with grant number DE240101089, and FL is also supported by ARC with grant number DP230101540 and the NSF&CSIRO Responsible AI program with grant number 2303037. This work was supported by the

National Natural Science Foundation of China No. 62125201. We thank Xiao Yu for implementing Text-Fluoroscopy.

## ETHIC STATEMENT

Our work follows the ICLR Code of Ethics. All data used in our paper are anonymized, eliminating any potential privacy concerns. There is no human or animal subjects to be involved in our paper. During the experiments, we strictly followed the ICLR Code of Ethics and made sure that this paper would not cause bias or other ethical issues. The proposed tool is designed to be transparent and reproducible, and the code will be released for public use and supporting the open-source community.

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

# APPENDIX

## CONTENTS

## A    EMPIRICAL ESTIMATION OF $\sigma_{YZXZ}$

An empirical estimation of $\sigma_{YZXZ}$ (Bounliphone et al., 2016) is:

$$\sigma_{YZXZ} \approx \frac{1}{m(m-1)^2} \mathbf{e}^\top \tilde{K}_{\mathbf{zz}} \tilde{K}_{\mathbf{zz}} \mathbf{e} - \left( \frac{1}{m(m-1)} \mathbf{e}^\top \tilde{K}_{\mathbf{zz}} \mathbf{e} \right)^2 \tag{8}$$

$$- \left( \frac{1}{m(m-1)r} \mathbf{e}^\top \tilde{K}_{\mathbf{zz}} K_{\mathbf{xz}} \mathbf{e} - \frac{1}{m^2(m-1)r} \mathbf{e}^\top \tilde{K}_{\mathbf{zz}} \mathbf{e} \mathbf{e}^\top K_{\mathbf{xz}} \mathbf{e} \right)$$

$$- \left( \frac{1}{m(m-1)n} \mathbf{e}^\top \tilde{K}_{\mathbf{zz}} K_{\mathbf{yz}} \mathbf{e} - \frac{1}{m^2(m-1)n} \mathbf{e}^\top \tilde{K}_{\mathbf{zz}} \mathbf{e} \mathbf{e}^\top K_{\mathbf{xz}} \mathbf{e} \right)$$

$$+ \left( \frac{1}{mnr} \mathbf{e}^\top K_{\mathbf{yz}} K_{\mathbf{xz}} \mathbf{e} - \frac{1}{m^2 nr} \mathbf{e}^\top K_{\mathbf{yz}} \mathbf{e} \mathbf{e}^\top K_{\mathbf{xz}} \mathbf{e} \right),$$

where $\mathbf{e}$ is a vector of ones with a length equal to the number of samples from $\mathcal{Z}$, and $\tilde{K}_{\mathbf{zz}}$, $K_{\mathbf{yz}}$, and $K_{\mathbf{xz}}$ are kernel matrices (Gretton et al., 2012a). The elements of $\tilde{K}_{\mathbf{xx}}$ are defined as

$$[\tilde{K}_{\mathbf{xx}}]_{ij} = \begin{cases} [K_{\mathbf{xx}}]_{ij} & \text{if } i \neq j, \\ 0 & \text{if } i = j. \end{cases}$$

Similar definitions apply to $\tilde{K}_{\mathbf{yy}}$ and $\tilde{K}_{\mathbf{zz}}$.

## B    MORE DETAILS FOR EXPERIEMENT SETTINGS

### B.1    DATASETS INTRODUCTION

We here use HC3 (human ChatGPT comparison corpus) (Guo et al., 2023), which contains $24,321$ paired answers from human and ChatGPT (OpenAI, 2022) with both long and short-level corpus; TruthfulQA (He et al., 2023; Lin et al., 2022), which comprises 817 questions from human (we here use best human answer), ChatGPT (OpenAI, 2022) and GPT4 (Anand et al., 2023). RAID (Dugan et al., 2024) is one of the largest MGT detection benchmark. It includes over 6 million generations spanning 11 models, 8 domains, 11 adversarial attacks and 4 decoding strategies. DetectRL (Wu et al., 2024) is another recent released MGT detection benchmark. It comprises 100,800 human-written samples, including 11,200 raw samples and 89,600 samples modified via attack manipulations. Additionally, it contains 134,400 samples generated by LLMs, categorized as follows: 11,200 samples generated with direct prompt, 22,400 with prompt attacks, 33,600 with paraphrase attacks, 33,600 with perturbation attacks, and 22,400 with data mixing.

### B.2    DATA SHUFFLE FOR DETECTION IN SECTION 4.2 AND SECTION 4.3

During each round of detection in section 4.2 and section 4.3, we first shuffle the HC3 dataset and select the first 512 tokens from HWTs and the first 512 tokens from MGTs as the text to be tested (the token number will be 256 in the token-256 experiments). The default reference data will be the rest of the data. We also test reference data with the same length as the text to be tested, *i.e.,* 512 or 256 tokens. For each experiment, the dataset will be as follows:

- In the HC3→HC3 experiment, we run our method and all baselines and save their detection result for this shuffle at this round. Specifically, we will shuffle 10 times in each round, where 5 shuffles select the text to be tested from HWT and the other 5 shuffles select the text to be tested from MGT. We have 10 rounds in total. Namely, our dataset is $10 \times 10$ for detection on 512-token texts and another $10 \times 10$ for detection on 256-token texts.

- In the HC3→TQA-ChatGPT experiment, we select 512 tokens (or 256 in the token-256 experiments) from TQA-ChatGPT as the text to be tested. We run our method and all baselines and save their detection result for this shuffle at this round. Specifically, we will shuffle 10 times in each round, where 5 shuffles select the text to be tested from HWT and the other 5 shuffles select the text to be tested from MGT. We have 10 rounds in total. Namely, our dataset is $10 \times 10$ for detection on 512-token texts and another $10 \times 10$ for detection on 256-token texts.

- In the HC3→TQA-GPT4 experiment, we select 512 tokens (or 256 in the token-256 experiments) from TQA-GPT4 as the text to be tested. We run our method and all baselines and save their

detection result for this shuffle at this round. Specifically, we will shuffle 10 times in each round, where 5 shuffles select the text to be tested from HWT and the other 5 shuffles select the text to be tested from MGT. We have 10 rounds in total. Namely, our dataset is $10 \times 10$ for detection on 512-token texts and another $10 \times 10$ for detection on 256-token texts.

## B.3   DATA SHUFFLE FOR DETECTION IN SECTION 4.5

Unless otherwise noted, we use the same way as the above section to choose the reference data. Namely, our reference data is always from HC3 dataset.

## B.4   INTRODUCTION OF BASELINES

We compare R-Detect to **five non-parametric baselines** and **seven parametric baselines**. The non-parametric baselines include two variations of our method 1) R-**Detect-**$k^m$: R-Detect with the Gaussian kernel optimised by median heuristic bandwidth (Bounliphone et al., 2016) and 2) R-**Detect w/o** $k^*$, namely our method without kernel optimisation.

The non-parametric baselines include 3) **MPP-HWT**, the MPP detector using HWT as the reference data, 4) **MPP-MGT**, is the MPP detector using MGT as as the reference data, and 5) **MPP-R**, a method that we proposed based on MPP, which uses both MGT and HWT for MPP's reference. The detection result will be given based on the hypothesis test with a smaller p-value from these two.

The parametric baselines include 6) **DetectGPT** (Mitchell et al., 2023), 7) **Radar** (Hu et al., 2023), 8) **Fast-DetectGPT** (Bao et al., 2024), 9) **DNA-GPT** (Yang et al., 2024a), 10) **Bino**, the state-of-the-art parametric detector (Hans et al., 2024) using 0.5 as the classification threshold; 11) **Bino-FPR**, the state-of-the-art parametric detector (Hans et al., 2024) using the threshold that especially for a low false positive rate, 12) **DALD** (Zeng et al., 2024), and 13) **Text-Fluoroscopy** (Yu et al., 2024).

## B.5   IMPLEMENTATION DETAILS OF OUR METHOD

The deep kernel $k_\omega$ in R-Detect is a neural network $\phi$ equipped with a feature extractor $g$. We learn the best kernel from HC3 data and used it for all test on HC3, TQA-ChatGPT and TQA-GPT4. The kernel is also used for MPP implementation to ensure a fair comparison.

Following the setting from (Zhang et al., 2024), the feature extractor $g$, we employ OpenAI's RoBERTa-based GPT-2 detector model (Liu, 2019) and consider its last hidden state as the feature of the input text. Each token in this feature extractor has a dimension of 768, and we set a maximum of 100 tokens per sentence. The network $k_\omega$ consists of a hidden-layer transformer followed by a projector and a multi-layer perceptron (MLP), where the projector reduces the data dimension from 768 to 512, while the MLP reduces the flattened data dimension from $51,200$ to 300. The data dimension during the whole procedure when feeding a sentence into the kernel follows the sequence: $100 \times 768 \rightarrow 100 \times 512 \rightarrow 51,200 \rightarrow 300$. Note that we only optimize the network $\phi$ and fix the mapping function $g$ during training through all our experiments.

We conduct our experiments using Python 3.9 and Pytorch 2.0 on a server with Intel Core i9 14900K and RTX 4090. In Algorithm 3, we use Adam optimizer (Kingma & Ba, 2015) to optimize the deep kernel parameters, we set $\lambda$ to $10^{-8}$ and batch size to 200, and the learning rate to 0.00005 in all experiments. The default threshold of the hypothesis test–both two-sample test or relative test—is $\alpha = 0.05$ to determine whether to reject or accept the null hypothesis. We also give the result when using different $\alpha$.

## B.6   IMPLEMENTATION DETAILS OF BASELINES

- MPP-HWT, MPP-MGT, and MPP-R: For the non-parametric detectors, MPP-HWT, MPP-MGT, and MPP-R, we applied the default parameter used in (Zhang et al., 2024) for HC3 data, which is the same with the setting of our method in section B.5.

- Bino, Bino-FPR: For the parametric detectors Bino and Bino-FPR, we use the default settings as the authors suggested, namely using Falcon-7B and Falcon-7B-Instruct models for scoring, the classification threshold 0.5 for Bino, and 0.8536432310785527 for Bino-FPR.

- R-**Detect-**$k^m$: R-Detect with the Gaussian kernel optimised by median bandwidth (Bounliphone et al., 2016) where the kernel bandwidth is from median heuristic.

- R-**Detect w/o** $k^*$: We use a Gaussian kernel with its width $\sigma = 1$

Table 5: Test power ($p$) and AUROC results on texts to be tested from HC3-MGT and HC3-HWT ($\alpha = 0.9$).

| Non-parametric Detectors | $tp^{\text{MGT}}$ | $tp^{\text{MGT}}_{\alpha=0.9,t=256}$ | $tp^{\text{HWT}}$ | $tp^{\text{HWT}}_{\alpha=0.9,t=256}$ | AUROC | $\text{AUROC}_{\alpha=0.9}$ |
|---|---|---|---|---|---|---|
| MPP-HWT | $1.00_{\pm0.00}$ | $1.00_{\pm0.00}$ | $0.10_{\pm0.12}$ | $0.10_{\pm0.12}$ | $0.72_{\pm0.26}$ | $0.72_{\pm0.26}$ |
| MPP-MGT | $0.90_{\pm0.15}$ | $0.90_{\pm0.10}$ | $0.00_{\pm0.00}$ | $0.00_{\pm0.00}$ | $0.72_{\pm0.24}$ | $0.72_{\pm0.24}$ |
| MPP-R | $1.00_{\pm0.00}$ | $1.00_{\pm0.00}$ | $0.01_{\pm0.03}$ | $0.01_{\pm0.03}$ | $0.99_{\pm0.01}$ | $0.99_{\pm0.01}$ |
| R-Detect(Ours) | $\mathbf{1.00}_{\pm0.00}$ | $\mathbf{1.00}_{\pm0.00}$ | $0.00_{\pm0.00}$ | $\mathbf{0.00}_{\pm0.00}$ | $\mathbf{1.00}_{\pm0.00}$ | $\mathbf{1.00}_{\pm0.00}$ |

Table 6: Test power ($p$) and AUROC results on texts to be tested from HC3-MGT and HC3-HWT (token size of 256 )(↑and ↓are compared to the default setting).

| Non-parametric Detectors | $tp^{\text{MGT}}$ | $tp^{\text{MGT}}_{t=256}$ | $tp^{\text{HWT}}$ | $tp^{\text{HWT}}_{t=256}$ | AUROC | $\text{AUROC}_{t=256}$ |
|---|---|---|---|---|---|---|
| MPP-HWT | $1.00_{\pm0.00}$ | $1.00_{\pm0.00}$ | $0.10_{\pm0.12}$ | $0.10_{\pm0.12}$ | $0.72_{\pm0.26}$ | $0.75_{\pm0.22}$ ↑ |
| MPP-MGT | $0.90_{\pm0.15}$ | $0.90_{\pm0.10}$ | $0.00_{\pm0.00}$ | $0.00_{\pm0.00}$ | $0.72_{\pm0.24}$ | $0.68_{\pm0.24}$ ↓ |
| MPP-R | $1.00_{\pm0.00}$ | $1.00_{\pm0.00}$ | $0.01_{\pm0.03}$ | $0.02_{\pm0.03}$↓ | $0.99_{\pm0.01}$ | $0.98_{\pm0.01}$ ↓ |
| R-Detect(Ours) | $\mathbf{1.00}_{\pm0.00}$ | $\mathbf{1.00}_{\pm0.00}$ | $0.00_{\pm0.00}$ | $\mathbf{0.00}_{\pm0.00}$ | $\mathbf{1.00}_{\pm0.00}$ | $\mathbf{1.00}_{\pm0.00}$ |

Table 7: Test power ($p$) and AUROC results on texts to be tested from HC3-MGT and HC3-HWT ($\alpha = 0.9, t = 256$) (↑and ↓are compared to the default setting).

| Non-parametric Detectors | $tp^{\text{MGT}}_{\alpha=0.9,t=256}$ | $tp^{\text{MGT}}_{\alpha=0.9,t=256}$ | $tp^{\text{HWT}}$ | $tp^{\text{HWT}}_{\alpha=0.9,t=256}$ | AUROC | $\text{AUROC}_{\alpha=0.9,t=256}$ |
|---|---|---|---|---|---|---|
| MPP-HWT | $1.00_{\pm0.00}$ | $1.00_{\pm0.00}$ | $0.10_{\pm0.12}$ | $0.10_{\pm0.12}$ | $0.72_{\pm0.26}$ | $0.75_{\pm0.22}$ ↑ |
| MPP-MGT | $0.90_{\pm0.15}$ | $0.90_{\pm0.10}$ | $0.00_{\pm0.00}$ | $0.00_{\pm0.00}$ | $0.72_{\pm0.24}$ | $0.68_{\pm0.24}$ ↓ |
| MPP-R | $1.00_{\pm0.00}$ | $1.00_{\pm0.00}$ | $0.01_{\pm0.03}$ | $0.01_{\pm0.03}$ | $0.99_{\pm0.01}$ | $0.98_{\pm0.01}$ ↓ |
| R-Detect(Ours) | $\mathbf{1.00}_{\pm0.00}$ | $\mathbf{1.00}_{\pm0.00}$ | $0.00_{\pm0.00}$ | $\mathbf{0.00}_{\pm0.00}$ | $\mathbf{1.00}_{\pm0.00}$ | $\mathbf{1.00}_{\pm0.00}$ |

## C  Non-parametric Detectors with Varied Settings

We here present the experiments of using the non-parametric detectors, especially MPP-HWT, MPP-MGT, MPP-R in different experimental settings such as 1) changing the default $\alpha = 0.05$ to $\alpha = 0.9$ ; 2) changing the token size from 512 to 256; 3) limiting the length of reference data from all available HC3 expect for the test to the same length with token size.

The result of 1) is presented in Table 5, with left column in each combined columns the default setting of $\alpha = 0.05$. Similarly, we have result 2) in Table 6 as well as the result for cross changes of both in Table 7. In addition, we also shorten the length of the reference data, decreasing it from the rest of HC3 to a portion that has the same length with the text to be tested. Namely, in the experiments of 512 the token number, we use two HWT and MGT within 512 tokens as the reference for the hypothesis testing. While in the experiments of 512 the token number, we use two HWT and MGT within 256 tokens as the reference for the hypothesis testing

## D  Toy Study: The Impact of Overlap Ratio

As noted in section 2.1, we consider only the case where texts belong exclusively to either $\mathcal{S}$h or $\mathcal{S}$m, given that $f$ is assumed to be a surjective function. However, in hypothesis testing such as the relative test, it is possible for $\mathcal{S}$h $\cap$ $\mathcal{S}$m $\neq \emptyset$. Therefore, it is important to examine the test power (i.e., the probability of rejecting the null hypothesis when the alternative hypothesis is true) in cases where $\mathcal{S}_h$ and $\mathcal{S}_m$ overlap. To investigate this, we conduct a toy study by manually **increasing the distance** $\mu$ between the distributions of $\mathbb{P}$ and $\mathbb{Q}$ i.e., **the mathematical reflection of decreasing the overlap ratio**. The results, presented in Table 9, indicate that the test power increases as the overlap between $\mathcal{S}_h$ and $\mathcal{S}_m$ decreases.

Table 8: Test power ($p$) and AUROC results when limits reference length to token size (↑and ↓are compared to the default setting).

| Non-parametric Detectors (↑↓compared to default setting) | $tp_r^{\mathrm{MGT}}$ | $tp_r^{\mathrm{MGT}}$ | $tp^{\mathrm{HWT}}$ | $tp_r^{\mathrm{HWT}}$ | AUROC | $\mathrm{AUROC}_r$ |
|---|---|---|---|---|---|---|
| MPP-HWT | $1.00_{\pm 0.00}$ | $1.00_{\pm 0.00}$ | $0.10_{\pm 0.12}$ | $0.27_{\pm 0.43}$ ↓ | $0.72_{\pm 0.26}$ | $0.72_{\pm 0.25}$ |
| MPP-MGT | $0.90_{\pm 0.15}$ | $0.93_{\pm 0.16}$ ↑ | $0.00_{\pm 0.00}$ | $0.02_{\pm 0.04}$ ↓ | $0.72_{\pm 0.24}$ | $0.72_{\pm 0.24}$ |
| MPP-R | $1.00_{\pm 0.00}$ | $1.00_{\pm 0.00}$ | $0.01_{\pm 0.03}$ | $0.01_{\pm 0.03}$ | $0.99_{\pm 0.01}$ | $0.98_{\pm 0.04}$ ↓ |
| R-Detect(Ours) | $\mathbf{1.00}_{\pm 0.00}$ | $\mathbf{1.00}_{\pm 0.00}$ | $0.00_{\pm 0.00}$ | $\mathbf{0.00}_{\pm 0.00}$ | $\mathbf{1.00}_{\pm 0.00}$ | $\mathbf{1.00}_{\pm 0.00}$ |

Table 9: Test power $p$ variations over distribution distance

| $\mu$ | 0.20 | 0.22 | 0.24 | 0.26 | 0.28 | 0.3 | 0.32 | 0.34 | 0.36 | 0.38 | 0.40 |
|---|---|---|---|---|---|---|---|---|---|---|---|
| $p$ | 0.83 | 0.82 | 0.82 | 0.83 | 0.86 | 0.89 | 0.92 | 0.95 | 0.97 | 0.98 | 0.99 |

# E EXPERIMENTAL ANALYSIS ACROSS DIVERSE BASELINES AND BENCHMARKS

## E.1 ENGLISH TEXT

We conducted more experiments across diverse baselines and benchmarks, and the detection results are shown in Table 10. The results show good performance of R-Detect over different benchmarks and under a variety of random added attacks. In these experiments, R-Detect is conducted in a **zero-shot setting**, namely using the same kernel and reference data from HC3 without further fine-tuning. Two sets of the token size have been picked, #256 and #512, noting that DetectGPT cannot run in the token#512 setting. The data shuffle for each column results is as follows:

- ***RAID-English***: we here only test on the English data in RAID, considering ***all model options*** in [*ChatGPT, GPT-4, GPT-3 (text-davinci-003), GPT-2 XL, Llama 2 70B (Chat), Cohere, Cohere (Chat), MPT-30B, MPT-30B (Chat), Mistral 7B, Mistral 7B (Chat)*]; and ***all adversarial attack options*** in [*Article Deletion, Homoglyph, Number Swap, Paraphrase, Synonym Swap, Misspelling, Whitespace Addition, Upper-Lower Swap, Zero-Width Space, Insert Paragraphs, Alternative Spelling*].

  During the data shuffles, we first use random seeds (Table 12) to shuffle the model options and attack options. Given the shuffled results, we randomly pick text from RAID with the shuffled settings. More specifically, each combination for the test data is as follows:

  - **mix1** represents a mixture of MGTs generated by *Mistral 7B (Chat)*, *GPT-2 XL*, *MPT-30B*, *MPT-30B (Chat)*, and *GPT-4*. without attacks;

Table 10: AUROC results on RAID (Dugan et al., 2024) and DetectRL (Wu et al., 2024) benchmarks over diverse baselines (the best detection performance is highlighted in **bold**.)

| Detectors | RAID-English [ACL'24] | | | | | | DetectRL-English [NeurIPS'24] | | | | |
|---|---|---|---|---|---|---|---|---|---|---|---|
| (token #256) | mix1 | mix2 | mix3 | att1 | att2 | att3 | non-att | h-att1 | h-att2 | p-att1 | p-att2 |
| MPP [ICLR24] | 0.7379 | 0.7032 | 0.8504 | 0.7104 | 0.5646 | 0.6952 | 0.6917 | 0.8021 | 0.7804 | 0.6374 | 0.6199 |
| DetectGPT [ICML23] | 0.6437 | 0.6632 | 0.4987 | 0.5931 | 0.5111 | 0.4554 | 0.5319 | 0.9656 | 0.6631 | 0.4500 | 0.5031 |
| Radar [NeurIPS23] | 0.8512 | 0.8124 | 0.7592 | 0.7988 | **0.5980** | 0.5964 | 0.8700 | 0.8328 | 0.7016 | 0.6964 | 0.9068 |
| Fast-DetectGPT [ICLR24] | 0.8400 | 0.8104 | 0.6320 | 0.4356 | 0.4124 | 0.3936 | 0.8768 | 0.9708 | 0.7296 | 0.6652 | 0.9024 |
| DALD [NeurIPS24] | 0.8284 | 0.7296 | 0.7872 | 0.7088 | 0.5324 | 0.5996 | 0.9032 | 0.8484 | 0.8196 | 0.9032 | 0.9420 |
| DNA-GPT [ICLR24] | 0.8108 | 0.3616 | 0.3176 | 0.7144 | 0.2064 | 0.3944 | 0.6708 | 0.8428 | 0.7780 | 0.9064 | 0.7088 |
| Text-Fluoroscopy [EMNLP24] | 0.9780 | 0.7196 | 0.7974 | 0.7664 | 0.5503 | 0.6771 | **1.0000** | 0.9812 | 0.8306 | **1.0000** | **1.0000** |
| R-Detect[ours] | **0.9996** | **0.8240** | **0.8808** | **0.9528** | 0.5684 | **0.7376** | 0.9964 | **0.9820** | **0.8692** | 0.9952 | **1.0000** |
| (token #512) | mix1 | mix2 | mix3 | att1 | att2 | att3 | non-att | h-att1 | h-att2 | p-att1 | p-att2 |
| MPP[ICLR24] | 0.6919 | 0.6531 | 0.7431 | 0.6409 | 0.6497 | 0.5671 | 0.6750 | 0.7490 | 0.7541 | 0.6741 | 0.5533 |
| DetectGPT [ICML23] | - | - | - | - | - | - | - | - | - | - | - |
| Radar [NeurIPS23] | 0.8948 | 0.9792 | 0.8792 | 0.8268 | 0.7432 | 0.7384 | 0.9720 | 0.6992 | 0.7916 | 0.9916 | **0.9732** |
| Fast-DetectGPT[ICLR24] | 0.8276 | 0.9704 | **0.9972** | 0.9092 | 0.9904 | 0.8180 | **1.0000** | **1.0000** | 0.8408 | **1.0000** | 0.8528 |
| DALD[NeurIPS24] | 0.8400 | 0.9792 | 0.9244 | 0.7284 | 0.8568 | 0.9068 | 0.9828 | 0.9492 | 0.7364 | 0.9764 | 0.5832 |
| DNA-GPT[ICLR24] | 0.7764 | 0.6388 | 0.6288 | 0.7608 | 0.7960 | 0.7412 | 0.8396 | 0.7664 | 0.7376 | 0.7844 | 0.8672 |
| Text-Fluoroscopy[EMNLP24] | 0.9824 | 0.8704 | 0.7728 | 0.8416 | 0.8164 | 0.4604 | 0.9936 | 0.9546 | **0.9632** | 0.9804 | 0.8976 |
| R-Detect[ours] | **1.0000** | **0.9688** | 0.9944 | **0.9908** | **0.9996** | **0.9184** | **1.0000** | 0.9996 | 0.8208 | **1.0000** | 0.9596 |

Table 11: AUROC results on RAID-Non English texts over non-parametric baselines (↑and ↓are compared to the default setting of zero-shot using token #256).

| Detector (token #256) | setting | N/Emix1 | N/Eatt1 | N/Emix2 | N/Eatt2 |
|---|---|---|---|---|---|
| MPP | zero-shot | 0.4717 | 0.4321 | 0.4496 | 0.3878 |
| R-Detect[ours] | zero-shot | 0.3048 | 0.5616 | 0.3504 | 0.4476 |
| MPP | new kernel | 0.5581↑ | 0.5426↑ | 0.4665↑ | 0.4316↑ |
| R-Detect[ours] | new kernel | 0.6052↑ | 0.7818↑ | 0.4272↓ | 0.3648↓ |
| MPP | new (kernel + reference) | 0.6141↑ | 0.4344↑ | 0.4682↑ | 0.5733↑ |
| R-Detect[ours] | new (kernel + reference) | 0.9490↑ | 0.8048↑ | 0.7704↑ | 0.5392↑ |
| **Detector (token #512)** | **setting** | **N/Emix1** | **N/Eatt1** | **N/Emix2** | **N/Eatt2** |
| MPP | zero-shot | 0.4879↑ | 0.5605↑ | 0.5605↑ | 0.4993↑ |
| R-Detect[ours] | zero-shot | 0.7770↑ | 0.5236↓ | 0.5236↑ | 0.5248↑ |

- **mix2** represents a mixture of MGTs generated by *Cohere*, *Cohere (Chat)*, *GPT-3 (text-davinci-003)*, and *GPT-2 XL* without attacks;
- **mix3** represents a mixture of MGTs generated by *MPT-30B*, *GPT-3 (text-davinci-003)*, *GPT-2 XL*, and *GPT-4*. without attacks;
- **att1** represents **mix1** under random attacks from all the attack options;
- **att2** represents **mix2** under random attacks from all the attack options;
- **att3** represents **mix3** under random attacks from all the attack options.

- *DetectRL-English*: DetectRL contains English texts only. We here test on the DetectRL, considering all LLMs and both ***human adversarial attacks*** by "*replacing one-quarter of the sentences in an LLM-generated text with human-written text at random*" or ***prompt attacks*** by "*intended to use carefully designed prompts to guide LLMs in generating text that closely mimics human writing style*":
  - **non-att**: represents a random sample from non-attacked DetectRL texts;
  - **h-att1** and **h-att2**: represents random samples from DetectRL texts under human adversarial attacks;
  - **p-att1** and **p-att2**: represents random samples from DetectRL texts under prompt attacks;

Table 12: Seed for Shuffle RAID (Dugan et al., 2024) and DetectRL (Wu et al., 2024)

| Seed | RAID-English [ACL'24] | | | | | | DetectRL-English [NeurIPS'24] | | | | |
|---|---|---|---|---|---|---|---|---|---|---|---|
| | mix1 | mix2 | mix3 | att1 | att2 | att3 | non-att | h-att1 | h-att2 | p-att1 | p-att2 |
| (token #256) | 5 | 3 | 2 | 5 | 3 | 2 | 0 | 0 | 5 | 4 | 0 |
| (token #512) | 5 | 1 | 0 | 5 | 0 | 1 | 3 | 3 | 6 | 3 | 7 |

## E.2 NON-ENGLISH TEXT

We use the non-English text from RAID to test R-Detect in the non-English case with and without attacks. Similar to the shuffle strategies in E.1, we shuffle the model and attack options before random sampling—**N/Eatt1** is the text of **N/Emix1** with attacks. We run the experiments under three settings:

- **zero-shot setting** is the default setting of using the kernel trained from HC3 and using HC3 data as the reference;
- **new kernel** refers to the setting that uses the non-English data to train a new kernel function and test it on unseen data. During the testing procedure, we still use HC3 shuffles as the reference;
- **new (kernel+reference)** refers to the setting that uses the non-English data to train a new kernel function and test it on unseen data. During the testing procedure, we shuffle samples from the non-English data used for training as the reference.

It can be concluded from comparing the results among settings that Although our method shows suboptimal performance in non-English detection tasks under a zero-shot setting, detection effectiveness improves significantly with an optimized kernel *e.g.* , increasing from 0.3048 to 0.6052 on

**N/Emix1**, and from $0.5616$ to $0.7818$ on **N/Emix1**. In addition, using proper references can further improve the detection effectiveness, noting that R-Detect reaches an AUROC of $0.9490$ on **N/Emix1** after using the non-English reference.

### E.3 ABLATION STUDY ON FEATURE EXTRACTOR

We run an ablation study on the RAID data, **mix1&2** and **att1&2**, by changing the feature extractor in MPP and R-Detect. The results are shown in Table 13. The detection performance of using falcon-rw-1b and roberta-base-openai-detector is better than the other feature extractors. This indicates a further improvement on non-parametric detectors by using more powerful feature extractors.

Table 13: Ablation study on feature extractor. The best feature extractor of each detector at the same setting is highlighted in **bold** (↑and ↓are compared to the default setting of roberta-base).

| Detector | Feature Extractor (token #256) | setting | mix1 | att1 | mix2 | att2 |
|---|---|---|---|---|---|---|
| MPP | roberta-base-openai-detector | new kernel | 0.6395↑ | **0.6044**↑ | 0.5791↓ | 0.6001 ↑ |
| | roberta-base | new kernel | 0.5791 | 0.5891 | 0.6221 | 0.5567 |
| | chatgpt-detector-roberta | new kernel | 0.7146↑ | 0.5997↑ | 0.6277↑ | 0.4105 ↓ |
| | falcon-rw-1b | new kernel | **0.7971**↑ | 0.5690↓ | **0.7749**↑ | **0.7059**↑ |
| R-Detect | roberta-base-openai-detector | new kernel | **0.9984**↑ | **0.9368**↑ | 0.9436 ↑ | 0.7868↑ |
| | roberta-base | new kernel | 0.9976 | 0.8576 | 0.9036 | 0.7862 |
| | chatgpt-detector-roberta | new kernel | 0.9960↓ | 0.7692↓ | 0.9696↑ | 0.6900↓ |
| | falcon-rw-1b | new kernel | 0.9808↓ | 0.8068 ↓ | **0.9852**↑ | **0.8632**↑ |

| Detector | Feature Extractor (token #512) | setting | mix1 | att1 | mix2 | att2 |
|---|---|---|---|---|---|---|
| MPP | roberta-base-openai-detector | new kernel | 0.8249↑ | 0.7587 ↑ | 0.7290↑ | 0.6283 |
| | roberta-base | new kernel | 0.7696 | 0.7412 | 0.7052 | **0.7402** |
| | chatgpt-detector-roberta | new kernel | 0.6571↓ | 0.5771 ↓ | **0.7754**↑ | 0.5906 ↓ |
| | falcon-rw-1b | new kernel | **0.9667**↑ | **0.8000**↑ | 0.5213↓ | 0.5723 ↓ |
| R-Detect | roberta-base-openai-detector | new kernel | **1.0000** ↑ | **1.0000**↑ | **0.9832** ↑ | **0.9884** ↑ |
| | roberta-base | new kernel | **1.0000** | 0.9700 | 0.8746 | 0.8656 |
| | chatgpt-detector-roberta | new kernel | 0.9932↓ | 0.9428↓ | 0.9432↑ | 0.7452↓ |
| | falcon-rw-1b | new kernel | **1.0000**↑ | **1.0000**↑ | 0.6472↓ | 0.7652↓ |

## F FIVE ESSAYS AND ITS RE-WRITTEN TEXTS BY GPT4O

We have randomly sampled five essays from the Essay dataset (Verma et al., 2024) and asked GPT-4o (OpenAI, 2024) to rewrite each of them by the prompt "rewrite" in Table 14.

We shuffle these human-written essays and GPT4o rewritten essays and take $512$ tokens from each as our test data in section 4.4. We here give each whole essay and its corresponding rewritten version from GPT4o.

Table 14: Five examples.

| Index | Human | GPT-4o |
|---|---|---|
| 1 | Robert Putnam makes the case that the United States has lost social capital and civil society over the last quarter-century after explaining the importance of social capital and civil society in democratic societies. Tocqueville during his visit to America said, ‚ÄúAmericans of all ages, all stations in life, and all types of disposition are forever forming associations. There are not only commercial and industrial associations in which all take part, but others of a thousand different types‚Äîreligious, moral, serious, futile, very general and very limited, immensely large and very minute‚Ä¶. Nothing, in my view, deserves more attention than the intellectual and moral associations in America‚Äù (Putmann, 2000, p. 65). Even though Putnam admits that determining the significance of a few countertrends without additional research is difficult, he concludes that critical factors such as social trust are eroding at an alarming rate in the United States. He then goes into great detail about this erosion, and he concludes by outlining the research required to investigate these theories thoroughly.

This important book demonstrates that social capital decreased dramatically in the United States after the mid-to-late-1960s due to generational succession, the rise of television and urban sprawl, and the increasing demands placed on people‚Äôs time and money, among other factors. Numerous social problems have been implicated from the same, and as a result, ranging from poor educational outcomes to economic stress and tensions between and among social groups. The most likely solution to this problem is to re-establish social capital in the manner of the Progressive Era but using contemporary approaches.

Societal norms of reciprocity and trustworthiness have developed due to the accumulation of social capital, defined as ‚Äúconnections among individuals‚Äô social networks.‚Äù Social capital is regarded as having the same value as physical and human capital... | Robert Putnam argues that the United States has experienced a significant decline in social capital and civil society over the past 25 years. He begins by emphasizing the importance of social capital and civil society in democratic societies. Tocqueville, during his visit to America, observed that "Americans of all ages, all stations in life, and all types of disposition are forever forming associations...Nothing, in my view, deserves more attention than the intellectual and moral associations in America" (Putnam, 2000, p. 65). While Putnam acknowledges the challenges in fully understanding the importance of countertrends without further research, he concludes that key elements like social trust are eroding at a concerning rate. He explores this erosion in detail and calls for research to investigate these trends further.

Putnam's work illustrates that social capital in the United States significantly declined after the mid-to-late 1960s due to factors such as generational change, the rise of television, urban sprawl, and increasing demands on people's time and resources. This decline is linked to various societal issues, including poorer educational outcomes, economic strain, and heightened tensions within and between social groups. Putnam suggests that revitalizing social capital, akin to efforts during the Progressive Era but with modern methods, could be a solution.

Social capital, defined as "connections among individuals' social networks," creates norms of reciprocity and trustworthiness, with value comparable to physical and human capital. In Bowling Alone, Putnam details the decline of social capital in contemporary America, examining its causes, consequences, and possible solutions.

Putnam distinguishes between two types of social capital: bonding capital, which binds individuals within a group, and bridging capital, which connects diverse social groups... |
| | Continued on next page | |

Continued

| Index | Human | GPT-4o |
|---|---|---|
| 2 | Archetypes describe such characteristics, which are universal patterns that define certain events, objects, or people. In literature, they describe representations and concepts that are common to all people, which allow people to understand and interpret stories. However, archetypes also exist in everyday life and play a significant role in shaping the perception of the world around them. In particular, there are archetypes of people that help determine their character and actions, as well as motivation.

Applying this concept to everyday life, one can single out special types of people who have a set of common and exclusive characteristics for them. They are formed both on the basis of the actions and inclinations of the person and depending on the environment. An example of an archetype in everyday life is the head of a large modern corporation, such as Elon Musk. He can be defined as a hero or creator who has the inspiration and courage to change the world around him. Other people perceive it in this way, which makes it a great example of an archetype. All his actions are motivated precisely by these characteristics, and his primary motivation is the transformation of the world. The presence of archetypes in everyday life allows me to compose a more comprehensive perception of people and events. In particular, based on the characteristics I know, I can determine the motivation and role of a certain person, which makes him more predictable. This is certainly more difficult than may be presented in the literature because people often have mixed features. However, in general, the concept of archetypes gives me the opportunity to understand the actions of people and their characters. I can later use this information to form my own relationship with them. | Archetypes represent universal patterns that define certain events, objects, or people. In literature, they reflect common themes and ideas that resonate across cultures, helping people interpret and relate to stories. However, archetypes are not limited to literature; they also exist in everyday life and play a vital role in shaping how we perceive the world. Specifically, archetypes of people help us understand their character, actions, and motivations.

In real life, we can identify certain types of individuals who share common traits shaped by both personal inclinations and their environments. For instance, a modern corporate leader like Elon Musk exemplifies the archetype of a hero or creator. His vision and courage to innovate make him a prime example of this archetype, as others often view him as someone driven by a desire to transform the world. These traits influence not only how he is perceived but also his actions and motivations.

Recognizing archetypes in daily life allows for a deeper understanding of people and events. By identifying these characteristics, I can better grasp someone's motivations and role, making their behavior more predictable. While real-life individuals may exhibit a blend of archetypal traits, the concept still helps me interpret their actions and personalities, which in turn informs how I interact with them. |

Continued

| Index | Human | GPT-4o |
|---|---|---|
| 3 | Introduction
The shifts in attitudes of people towards romantic relationships and, more specifically, marriage are attributes of societal development. From this perspective, the trends common for Gen Z are incredibly informative and, therefore, used for analyzing the situation in this area (Klein, 2022). It is clear that individuals‚Äô mindsets are formed under the influence of external circumstances, which implies the correlation of opinions about love and sex with the presence of threats to personal well-being. Hence, one cannot examine the accompanying phenomena and cause-and-effect relationships while neglecting either the human factor or the conditions of the environment. One of the articles devoted to the subject is ‚ÄúAre Gen Z more pragmatic about love and sex?‚Äù and its critical analysis can reveal the elements of effective writing intended for considering complex events.
Summary of the Article
In the selected piece, Jessica Klein narrates the tendency for representatives of Gen Z to postpone marriage and avoid long-term relationships while prioritizing personal issues. Compared to previous generations, these individuals do not view this area as a natural condition for entering adulthood due to a number of reasons. First, the advantages of single life are more attractive than settling down and help them satisfy their need for introspection (Klein, 2022). Second, people in this group adopt a pragmatic approach to the matter because they understand the necessity of gaining stability in life in order to have a family (Klein, 2022). Third, Gen Zers face more problems than their parents due to the pandemic and prefer to reflect on the kind of person they want to be partners with and choose them carefully (Klein, 2022). As a result, the processes in this area slow down significantly, and the primary focus on self, as opposed to interactions with others, justify the changes which happened to this part of human life... | Introduction
Shifts in attitudes toward romantic relationships, particularly marriage, are closely tied to societal development. In this context, trends observed among Gen Z are especially informative and serve as a basis for analyzing changes in this area (Klein, 2022). It is evident that people's mindsets are shaped by external factors, meaning opinions about love and sex are influenced by conditions that affect personal well-being. As a result, it is impossible to examine the underlying causes and relationships without considering both human factors and environmental conditions. One relevant article addressing this topic is "Are Gen Z More Pragmatic About Love and Sex?" and its critical analysis highlights key elements of effective writing used to discuss complex social phenomena.
Summary of the Article
In the article, Jessica Klein explores the tendency of Gen Z to delay marriage and avoid long-term relationships, placing greater emphasis on personal priorities. Unlike previous generations, Gen Z does not view romantic relationships as a natural step into adulthood, driven by several factors. First, they find the benefits of single life more appealing than settling down, which allows them to focus on self-reflection (Klein, 2022).
Second, many Gen Z individuals take a pragmatic approach, recognizing the importance of achieving personal stability before starting a family (Klein, 2022). Third, the pandemic has brought additional challenges to their lives, prompting them to carefully consider the kind of partners they seek (Klein, 2022). As a result, the pace of romantic developments has slowed, with an emphasis on personal growth taking precedence over interpersonal relationships, reflecting broader changes in modern life... |

Continued

| Index | Human | GPT-4o |
|-------|-------|--------|
| 4 | From 2016 to 2018, this study summarizes sport fisheries in the Anchorage Management Area. They contained a description and historical background of each sport fishery, as well as information on how each fishery is now managed, as well as performance and escapement data for 2016‚Äì2018. According to the report, reductions in the number and size of fish available for release in locally stocked lakes have had a detrimental impact on local fisheries‚Äô popularity and productivity. It should be mentioned that because this source gives statistical data, it may be trusted. All of this will surely be beneficial to the study, as official data must be considered.

The sport fisheries in the North Gulf Coast Management Area are discussed in this study. A summary and overview of each fishery are included, and information on how the fisheries are managed and fishery performance and escapement from 2016 to 2018. This report is provided by the official department that controls fishing activities in Alaska. However, it should be noted that this source can be considered reliable as it provides statistical information. All this will undoubtedly be useful for research since it is necessary to take into account official data.

The intricacy of variables that restrict or encourage diversity in Alaskan fisheries, one of the world‚Äôs most important fishing locations, is examined in this case study. They looked at changes in participation and portfolio variety in Alaskan fisheries over three decades using a mix of harvest statistics and literature reviews. The four case studies looked at how fishers, fleets, and communities responded to various interconnected stressors, such as biological decreases, market and pricing dynamics, and fisheries privatization. Experts also verified this article, which is reflected in the reliability and accuracy of the information provided. It will be useful for research because it reflects current trends and factors affecting the spread of sport fishing. | From 2016 to 2018, this study summarizes the sport fisheries in the Anchorage Management Area, providing descriptions, historical background, and details on how each fishery is managed. It also includes performance and escapement data for the years 2016–2018. According to the report, reductions in the number and size of fish stocked in local lakes have negatively impacted the popularity and productivity of these fisheries. As this source provides statistical data, it is considered reliable and will be highly valuable to the study, as official data must be taken into account.

The sport fisheries in the North Gulf Coast Management Area are also discussed in this study. It includes an overview of each fishery, management practices, and fishery performance and escapement data from 2016 to 2018. This report is issued by the official department responsible for regulating fishing activities in Alaska, and since it provides statistical information, it can be trusted as a reliable source. The data will be essential for the research, as incorporating official data is crucial.

A case study is presented on the complexity of factors that either limit or promote diversity in Alaskan fisheries, one of the world's most significant fishing regions. This study examines changes in participation and portfolio diversity in Alaskan fisheries over three decades, using a combination of harvest data and literature reviews. Four case studies explore how fishers, fleets, and communities responded to interconnected challenges such as biological declines, market dynamics, pricing, and the privatization of fisheries. The article has been verified by experts, ensuring the reliability and accuracy of the information. It will be valuable for the research, as it reflects current trends and factors influencing sport fishing. |

Continued

| Index | Human | GPT-4o |
|---|---|---|
| 5 | Deductive reasoning involves the construction and development of cause-and-effect solid relationships. It is noteworthy that the connection is established either between two facts or a point and the idea of how it will affect the future (Besnard & Hunter, 2018). Thus, a judgment (logical expression) includes the first ‚Äì the prerequisites, the second ‚Äì the conclusion. In logic, an argument is valid only when it has a form that makes it impossible for the premises to be correct and the judgment to be wrong. According to the explained three types of deductive arguments, a valid statement does not need to have actually trustworthy premises. Still, it must have ones that, if they were accurate, would guarantee that the conclusion is proper. Example: 1. All metals are heavy 2. Tin is a metal 3. Tin is heavy A sound argument must have both a valid form and true premises. Valid arguments can be unsound, but they will have wrong assumptions. Example: 1. All dogs are cats 2. All mice are dogs 3. All mice are cats What makes this a valid statement is not that it has trustworthy premises and a proper conclusion but the logical necessity of the judgment, given the two assumptions. The argument would be just as valid if the premises and conclusion were wrong. However, it is essential to determine that the purpose of argumentation is to establish the truth of a judgment. An invalid argument does not achieve this goal, so it cannot be considered sound (Besnard & Hunter, 2018). A sound argument has a valid form and true premises; thus, it cannot be invalid in all cases. If the argument is invalid, then the beliefs thoroughly do not matter, and, in fact, only the conclusion remains. Therefore, this means that there is no argument in this case, especially a sound one. | Deductive reasoning involves constructing and establishing strong cause-and-effect relationships. These connections are typically drawn between two facts or between an idea and how it may impact the future (Besnard & Hunter, 2018). In deductive logic, an argument consists of two parts: the premises and the conclusion. A valid argument is one where, if the premises are true, it would be impossible for the conclusion to be false. However, the premises of a valid argument do not necessarily have to be true. The key is that if they were true, they would guarantee the correctness of the conclusion. For example: 1. All metals are heavy. 2. Tin is a metal. 3. Therefore, tin is heavy. A sound argument, on the other hand, requires both a valid structure and true premises. While valid arguments can be unsound if the premises are false, they still follow a logical structure. For example: 1. All dogs are cats. 2. All mice are dogs. 3. Therefore, all mice are cats. This is a valid argument because the conclusion logically follows from the premises, but it is not sound because the premises are false. The validity of the argument comes from the logical necessity of the conclusion given the assumptions, even if those assumptions are incorrect. However, the goal of argumentation is to establish the truth of a conclusion. An invalid argument does not meet this goal and cannot be considered sound (Besnard & Hunter, 2018). Therefore, a sound argument must always have both a valid form and true premises. If an argument is invalid, its premises are irrelevant, and there is no argument at all, much less a sound one. |

