# OpenReview forum: "Deep Kernel Relative Test for Machine-generated Text Detection"
_ICLR.cc/2025/Conference — ICLR 2025 Poster_

### Official Review · Reviewer_Er1T · 2024-10-29

**Soundness:** 3
**Presentation:** 2
**Contribution:** 2
**Rating:** 6
**Confidence:** 3

**Summary:**

This paper examines the issue of how to detect machine-generated texts. It suggests that previous methods are unable to identify human-written texts when there is a change in distribution. It proposes to employ non-parametric kernel relative test to detect machine-generated texts by testing whether it is statistically significant that the distribution of a text to be tested is closer to the distribution of human-written texts than to the distribution of machine-generated texts. A kernel optimization algorithm is proposed to select the best kernel that can enhance the testing capability for machine-generated text detection. Some experiments support the effectiveness of the method proposed in this paper.

**Strengths:**

1. The motivation of the paper is reasonable, and the problem it addresses has practical application value.

2. The method proposed in the paper is simple and effective, with a coherent logic, and the description of the algorithm is very clear.

3. The paper has a solid theoretical foundation, its arguments are logically sound, and the method is highly interpretable.

4. The experiments indicate that the algorithm performs well.

**Weaknesses:**

1. The assumptions on which the method is based are too strong and do not align with reality. Firstly, there is significant overlap between machine-generated language and human language; they do not exist in completely separate domains. Additionally, the subspace assumption is overly idealized and lacks a solid foundation, which greatly undermines the paper's validity.

2. The method proposed in the article resembles a general text anomaly detection approach and is not closely related to large language models or machine-generated language detection. It appears to be a universal solution for text anomaly detection rather than a targeted one, as the specific characteristics of the problem have not been discussed or reflected in the method's design.

3. The comparative methods in the experiments are relatively few and not comprehensive enough. Many anomaly detection solutions could also address this issue, so more comparisons and discussions should be provided.

**Questions:**

Please see weaknesses.

---

> ### Author Response · Authors · 2024-11-24
>
> **We thank you for reviewing our paper and giving valuable comments. We will include rebuttal discussions in the updated version to make the final version more clear to our readers. We hope the answers below solve your concerns. If your concerns were not cleared, please let us know, and we'll try to clarify them further. We also hope you can re-evaluate our paper based on the answers and consider raising your score.**
>
> >Weakness 1: The assumptions on which the method is based are too strong and do not align with reality. Firstly, there is significant overlap between machine-generated language and human language; they do not exist in completely separate domains. Additionally, the subspace assumption is overly idealized and lacks a solid foundation, which greatly undermines the paper's validity.
>
> We would like to clarify that the purpose of introducing this assumption is to ensure the theoretical rigor of our proposed method. As you correctly pointed out, it is clearly true in the wild that there is significant overlap between machine-generated language and human language, and we fully acknowledge this fact. Consequently, it is **impossible** for **any detector** to reliably differentiate machine-generated texts (MGTs) from human-written texts (HWTs) within this overlapping area.
>
> This raises an important **question**: when a detector fails in practice, is the **failure due to inherent flaws** in its design, or is it simply **encountering this theoretically undetectable overlap?**
> By explicitly incorporating this assumption, we aim to delineate the boundary where our method—and indeed any method—might face principled limitations, thereby providing a clearer understanding of the contexts in which failure is expected.
>
> > Weakness 2: The method proposed in the article resembles a general text anomaly detection approach and is not closely related to large language models or machine-generated language detection. It appears to be a universal solution for text anomaly detection rather than a targeted one, as the specific characteristics of the problem have not been discussed or reflected in the method's design.
>
> The theoretical support of our method is **relative hypothesis test**, which is a universal solution to test if the hypothesis "the distribution **Z** is closer to the distribution **Q** than to the distribution **P**" is statistically true. Originally, it is a **universal solution** at the distribution level **rather than one targeted at MGT detection**.
>
> However, **no existing study explicitly explores whether or how relative testing** can be used for MGT detection. Therefore, one of **our key contributions is to fill this gap**. The experimental results of R-Detect-**k^m** and R-Detect w/o **k\*** in Table 2 and Table 3 indeed show that **the original relative test cannot work very well in MGT detection tasks**, especially when the HWTs are from unseen distribution. Our proposed version R-Detect can overcome that and perform excellently in MGT detection tasks.
>
> In addition, the intuitive explanations of how the Test Power in relative hypothesis test is connected with TPR/FPR in MGT detecion can be found in Section 3.1 **Test Power *v.s.* TPR *v.s.* FPR.**  (Line #206, highlighted in cyan).
>
> > The comparative methods in the experiments are relatively few and not comprehensive enough. Many anomaly detection solutions could also address this issue, so more comparisons and discussions should be provided.
>
> - We are conducting more experiments. As one of our key contributions is ``enriching the MGT detection framework with robust theoretical foundations derived from hypothesis testing", we add more comparative experiments to existing MGT detection methodologies.
> Following reviewer `[4sdF]`'s comments the experiments will include
>
> 1. **More baselines** such as few-shot detection `[ICLR'24]`, threads-of-subtlety (awaiting code availability) `[ACL'24]`, Detect-GPT `[ICML'23]`, Fast-DetectGPT `[ICLR'24]`, DNA-GPT `[ICLR'24]`, DALD `[NeurIPS'24]`.
> 2. Testing on the benchmark proposed in RAID `[ACL'24]`, which includes data with **adversarial attacks** and **non-English texts.**
> 3. Testing **feature generators other than Roberta-based GPT-2**, such as ChatGPT Detector `[r1]`.
> `[r1]` *How close is ChatGPT to human experts? Comparison corpus, evaluation, and detection.* ArXiv 2023.
> - The results will be updated gradually in rebuttal replies and updated to the submission.

---

> > ### Comment · Reviewer_Er1T · 2024-11-25
> >
> > 1. The first issue concerns the logical structure of the paper. You stated, "the purpose of introducing this assumption is to ensure the theoretical rigor of our proposed method." This is not the correct approach to conducting research. You cannot derive assumptions based on the method and then design the problem around those assumptions. Assumptions should originate from the problem, and methods should serve these assumptions—not the other way around. If you create assumptions based on the method and then design the problem to align with these assumptions, it leads to a disconnect between the assumptions and the real problem. You could have an effective method first and then provide explanations to support it. Your current method could be explained in various ways, yet you chose to justify it using counterfactual assumptions. You may believe that this approach enhances the logical structure of the paper, but, in fact, deviating from reality is what undermines logic. No detector can completely differentiate machine-generated and human-written texts. I am not asking you to achieve that, but your assumption that they can be separated damages the rigor of the study. You could have avoided making this assumption while still deriving your method. Regarding the subspace assumption, it also serves your method, but you have not explained its connection to real-world problems. It is unclear under what circumstances such an assumption is valid and when your method would be applicable.
> >
> > 2. The logical issue here lies in your statement, "However, no existing study explicitly explores whether or how relative testing can be used for MGT detection. Therefore, one of our key contributions is to fill this gap." The correct logical approach should involve analyzing the real-world MGT detection problem first, identifying its specific characteristics, and then determining whether these characteristics align with the relative testing method. If such alignment is found, you can then justify using this method to solve the problem. However, your current logic appears to be that you noticed relative testing had not been applied to MGT detection, so you decided to use it. This approach results in a weak motivation for your study. First, you do not explain in your paper what distinctive characteristics set the MGT detection problem apart from other text detection or classification problems. Second, you do not clarify why these distinctive characteristics make relative testing an appropriate or particularly well-suited method for addressing this problem. As a result, your explanation of the research rationale lacks sufficient depth and persuasiveness.
> >
> > 3.  I will not change my score based on promised but unpublished experimental results.
> >
> > In summary, the overall argumentation of the paper is still insufficient. If the experimental results are sufficiently comprehensive and demonstrate strong performance, they might compensate for some of these shortcomings.

---

> ### Author Response · Authors · 2024-11-27
> **Replies to Comments 1,2**
>
> Thank you for your time and prompt replies!
> We apologise for potentially misunderstanding your questions in the first round. Below, we clarify our responses to the points you raised:
>
> ---
>
> #### 1. **Should the assumptions originate from the problem?**
>
> The key gap lies in how we interpret "assumption." If an "assumption" refers to a mathematical description of the problem, it must stem from the practical problem. In the original submission, we used **Problem 1** to describe this practical problem:
> *MGT detection aims to find a detector $f: S_h \cup \mathcal{S}_{\rm m} \rightarrow \{\text{MGT}, \text{HWT}\}$, which effectively distinguishes between MGTs and HWTs.*
> As you noted, the method can still be derived without this assumption. From the perspective of problem description, this is true, as the detection problem is fundamentally about finding a mapping function that maps text to MGT or HWT, regardless of whether MGTs and HWTs overlap.
>
> However, in our original submission, the assumption refers to the method itself. For instance, suppose the word "cat" is written by a human while another "cat" is generated by a machine. In this case, the text "cat" overlaps between HWTs and MGTs. If we assume that this text can belong to both HWT and MGT, the corresponding mathematical problem becomes "the same $X$ has two different $Y$s," which is a completely different story.
>
> Thanks again for your valuable insights on this point! We realise this point hasn't been explicitly explained in the main context. We will refine the draft with the following revisions:
>
> 1. Move the assumption after proposing Problem 1. Instead of "overlapped", it should be "identical" in the assumption.
> 2. Add an explanation of the assumption between Problem 1 and the assumption.
> 3. Include the following toy study as a figure to demonstrate how detection performance varies when we manually increase the distance  $\mu$   between the distributions of $\mathbb{P}$ and $\mathbb{Q}$ (i.e., the mathematical reflection of decreasing the overlap ratio). The results show that test power increases as the overlap between distributions decreases.
>
> | $\mu$       | 0.2  | 0.22 | 0.24 | 0.26 | 0.28 | 0.3  | 0.32 | 0.34 | 0.36 | 0.38 | 0.40 |
> |------------------|-------|------|------|------|------|------|------|------|------|------|------|
> | **Test Power**   | 0.83  | 0.82 | 0.82 | 0.83 | 0.86 | 0.89 | 0.92 | 0.95 | 0.97 | 0.98 | 0.99 |
>
> ---
>
> #### 2. **Weak Motivation**
>
> In the first round, one of your concerns was: *"It appears to be a **universal solution** for text anomaly detection rather than a targeted one."* That is why we responded by explaining that the relative test **is a universal solution** but has not yet been explored for MGT detection.
>
> In the MGT detection problem (our Problem 1), the effectiveness is evaluated using TPR/FPR. Intuitive explanations of how **Test Power in hypothesis testing** relates to TPR/FPR in MGT detection can be found in Section 3.1: *Test Power vs. TPR vs. FPR* (Line #206, highlighted in cyan):
> *"Consequently, the test power in R-Detect, given that the ground truth is an MGT, is the probability that an MGT is correctly identified as an MGT, corresponding to TPR. Similarly, the test power in R-Detect, given that the ground truth is an HWT, is the probability that an HWT is incorrectly identified as an MGT, corresponding to FPR."*
>
> Test power and hypothesis test have their statistical support. Therefore, as pointed in our first contribution:
> *"The potential of using statistical hypothesis tests for MGT detection is explored, enriching the detection framework with robust theoretical foundations derived from hypothesis testing."* (Line #102)

---

> > ### Author Response · Authors · 2024-11-27
> > **Replies to Comment 3**
> >
> > #### 3. **Experimental Results**
> >
> > The detection results for various methods/benchmarks are presented in the tables under `section: Newly Added Experiments: Result Summary, Design, and Notations` ([OpenReview link](https://openreview.net/forum?id=z9j7wctoGV&noteId=TISnKQzKS2)), using AUROC as the metric. For your convenience, we summarize the results below:
> >
> > 1. **Table r1**: Our method performs the best across all baselines on complex MGT detection tasks (using baselines and tasks suggested by Reviewer `[4sdF]`).
> > 2. **Table r2**: Our method is robust against adversarial attacks compared to other baselines (using baselines and tasks suggested by Reviewer `[4sdF]`).
> > 3. **Table r3**: While our method does not perform well in **non-English** detection tasks in a zero-shot setting, detection effectiveness improves significantly with a properly optimized kernel (e.g., from 0.3 to 0.61 on mix-1, and 0.35 to 0.43 on mix-2). It is worth noting that the **kernel is trained with English** RAID data.
> > 4. **Table r4**: In our default setting in the original submission, `roberta-base-openai-detector` is the feature extractor. We compared it with other available feature extractors and highlighted the best extractor for each dataset in bold. Results indicate that using a more powerful extractor, such as `falcon-rw-1b` (noting that Bino uses `falcon-7b`), further improves performance.
> >
> > ---
> >
> > #### Experiments in Progress
> >
> > 1. The TBC values in Table r3 will be gradually replaced.
> > 2. The same experiments are being conducted with a token size of 512.
> >
> > Once all these are complete, the experimental results and analysis will be added to the main draft and Appendix.
> >
> > If you have any additional questions or concerns, please let us know.

---

> > > ### Comment · Reviewer_Er1T · 2024-11-28
> > >
> > > Thank you for your rigorous approach and efforts in improving the experiments. I am inclined to believe that the authors will continue to refine this work in the future. I have updated my score to 6 and encourage the authors to keep enhancing their work.

---

> > > > ### Author Response · Authors · 2024-12-04
> > > > **Rebuttal Summary**
> > > >
> > > > Dear Reviewer Er1T,
> > > >
> > > > We sincerely thank you for your responsibility and for helping us develop a thorough discussion and improve our paper!
> > > >
> > > > During the rest of the rebuttal period, we have been working on the newly suggested experiments to further refine our work. All experiments using token#256 have now been completed. Below, we present a summary of the experimental analysis:
> > > >
> > > > ### Experiments Conducted
> > > >
> > > > #### Five Additional Baselines:
> > > > - **DetectGPT** [ICML'23]
> > > > - **Fast-DetectGPT** [ICLR'24]
> > > > - **DNA-GPT** [ICLR'24]
> > > > - **DALD** [NeurIPS'24]
> > > > - **Text Fluoroscopy** [EMNLP'24] (special thanks to the authors for their kind assistance in running the experiments)
> > > >
> > > >
> > > > Eventually, we didn't make the test on **Threads-of-Subtlety** [ACL'24] (requested by Reviewer 4sdF) because their code is not yet available.
> > > >
> > > > #### Three Additional Benchmarks:
> > > > - **RAID** [ACL'24]
> > > > - **Beemo** [arXiv'24]
> > > > - **DetectRL** [NeurIPS'24]
> > > >
> > > >
> > > > ### Summary of Results
> > > > The results are presented in Tables r1–r8:
> > > >
> > > > - **Table r1**: Our method achieves the best performance on complex MGT detection tasks.
> > > > - **Table r2**: Our method demonstrates robustness against adversarial attacks.
> > > > - **Table r3**: While our method initially underperforms in non-English detection tasks in a zero-shot setting, detection effectiveness improves significantly with a properly optimized kernel (e.g., from 0.3 to 0.61 on mix-1, and 0.35 to 0.43 on mix-2).
> > > > - **Table r4**: Using a powerful extractor, such as falcon-rw-1b (noting that Bino uses falcon-7b) can further enhance performance.
> > > > - **Table r5**: Demonstrates that our method outperforms other baselines on the **Beemo** [arXiv'24] benchmark.
> > > > - **Table r6**: Shows that our method achieves superior results on the **DetectRL** [NeurIPS'24] benchmark.
> > > > - **Tables r7–r8**: Confirms that our method surpasses **Text Fluoroscopy** [EMNLP'24] across the **RAID**, **Beemo**, and **DetectRL** benchmarks, both with and without attacks.
> > > >
> > > >
> > > >
> > > > We regret that some suggestions regarding assumptions have not been updated in the resubmission due to the deadline. However, we commit to revising and including the toy data test in the final version.
> > > >
> > > > All the best,
> > > > The Authors

---

> ### Author Response · Authors · 2024-11-28
> **Thanks for updating your score to 6!**
>
> Dear Reviewer Er1T,
>
> Thank you so much for taking the time to thoroughly review our paper and provide insightful feedback! We really enjoy the discussion with you and deeply appreciate your support!
>
> Best,

---

### Official Review · Reviewer_4sdF · 2024-10-30

**Soundness:** 2
**Presentation:** 2
**Contribution:** 2
**Rating:** 6
**Confidence:** 5

**Summary:**

This submission proposes to address the limitation of existing non-parametric LLM detector approach that tends to make mistakes in identifying HWTs that deviate from the
seen HWT distribution. The authors suggest to employ non-parametric kernel relative test to address the issue. Basically, the idea has some novelty to some extend. The paper highly follow the method in paper “DETECTING MACHINE-GENERATED TEXTS BY
MULTI-POPULATION AWARE OPTIMIZATION FOR
MAXIMUM MEAN DISCREPANCY” from both methodology style and the experimental design.

**Strengths:**

1). Basically, the idea has some novelty to some extend.

2). The problem to address is important.

3). The writting is somehow clear and easy to follow.

**Weaknesses:**

My major concerns are as follows:

1). The limitation of these MMD-based approach is obvious. The appraoch by nature is not Zero-Shot aaproach. As the approach needs to prepare HWT and MGT in advance, I would say the approach is not training free by nature. Especially, the paper proposes kernel optimisation algorithm to select best kernel. This by nature is somehow “training”. Thus, the approach only compare times with Bino is not sufficient, as it also needs to compare time with training/classifier based approach (Table 2)

2). The experiments are far from complete. Bascially, the datasets used are obviously too simple. Why not use the RAID datasets (in the following paper) that includes 22 different LLMs with different settings?
“RAID: A Shared Benchmark for Robust Evaluation of Machine-Generated Text Detectors, ACL’24”

3). Need to validate the robustness of the detection algorithm under adversial attack.

4). Need to provide non-english dattasets’ results to see the effectiveness on non-english settings.

5). As mentioned in 1), the paper needs to provide comparisons to other classification based approaches as well other metric-based/ logits-based approaches. Such as:
Classification-based:
FEW-SHOT DETECTION OF MACHINE-GENERATED TEXT USING STYLE REPRESENTATIONS. ICLR’24. https://arxiv.org/pdf/2401.06712
Threads of Subtlety: Detecting Machine-Generated Texts Through Discourse Motifs. ACL’24. https://arxiv.org/pdf/2402.10586.
Logits-based:
DetectGPT (ICML’23)
Fast-DetectGPT (ICLR’24)
DNA-GPT [ICLR’24]
DALD: Improving Logits-based Detector without Logits from Black-box LLMs. [NeurIPS’24]

6). The paper also highly depends on the feature extractor, that is “fine-tuned a RoBERTa model on GPT-2-generated”. What’s the performance for other feature extractors? (need experiements). Need to explain the major different and advantage of proposed approach over training-based approaches.

7). The literature review is far from sufficient. Pl refer to the recent surveys such as:
A Survey of AI-generated Text Forensic Systems: Detection, Attribution, and Characterization
A Survey on Detection of LLMs-Generated Content, EMNLP’24.
On the Possibilities of AI-Generated Text Detection

**Questions:**

refer to the weakness

**Details Of Ethics Concerns:**

No concerns

---

> ### Author Response · Authors · 2024-11-24
>
> **We thank you for reviewing our paper and giving valuable comments. We will include rebuttal discussions in the updated version to make the final version more clear to our readers. We hope the answers below solve your concerns. If your concerns were not cleared, please let us know, and we'll try to clarify them further. We also hope you can re-evaluate our paper based on the answers and consider raising your score.**
>
> >Concern 1): The limitation of these MMD-based approach is obvious. The appraoch by nature is **not Zero-Shot aaproach**. As the approach needs to prepare HWT and MGT in advance, I would say the approach is not training free by nature. Especially, the paper proposes kernel optimisation algorithm to select best kernel. This by nature is somehow “training”. Thus, the approach only compare times with Bino is not sufficient, as it also needs to compare time with training/classifier based approach (Table 2)
>
> - It is worth noting that although Bino is zero-shot, it is based on the output of pre-trained LLMs, while our "pre-training" is merely a deep kernel using an NVIDIA RTX graphics card with 16 GB of memory.
> - It is also worth noting that we only **trained the kernel once with HC3 and tested it for ALL experiments**. Fairly speaking, the experiments in **Section 4.2.2** (when HWTs are from unseen distributions) and **Section 4.4** (detecting GPT4-rewritten texts) can be considered **zero-shot** testing. Clearly, these are more challenging tasks.
>
> >Concerns 2)-6): we understand these concerns as insufficient experiments
>
> - We are conducting more experiments,where we use the **same kernel trained from HC3.** Following your comments, the experiments will include:
> 1. **More baselines** such as few-shot detection `[ICLR'24]`, threads-of-subtlety (awaiting code availability) `[ACL'24]`, Detect-GPT `[ICML'23]`, Fast-DetectGPT `[ICLR'24]`, DNA-GPT `[ICLR'24]`, DALD `[NeurIPS'24]`.
> 2. Testing on the benchmark proposed in RAID `[ACL'24]`, which includes data with **adversarial attacks** and **non-English texts.**
> 3. Testing **feature generators other than Roberta-based GPT-2**, such as ChatGPT Detector `[r1]` and Falcon-7B's feature generator (e.g., Bino's).
> `[r1]` *How close is ChatGPT to human experts? Comparison corpus, evaluation, and detection.* ArXiv 2023.
>
> In the above three experiments, we **directly used the previously trained kernel** from HC3 and **didn't train the kernel** with new data. Additionally, we will conduct
>
> 4. the effectiveness of our method when **training the kernel with RAID** data.
> - The results will be updated gradually in rebuttal replies and updated to the draft.
>
> > Concern 7): The literature review is far from sufficient. Pl refer to the recent surveys such as: A Survey of AI-generated Text Forensic Systems: Detection, Attribution, and Characterization A Survey on Detection of LLMs-Generated Content, EMNLP’24. On the Possibilities of AI-Generated Text Detection\\
>
> We have reviewed *``On the Possibilities of AI-Generated Text Detection"* in the first version (Line \#96, \#117, and \#477).
> We add the review of *``A Survey of AI-generated Text Forensic Systems: Detection, Attribution, and Characterization"* and *``A Survey on Detection of LLMs-Generated Content"* in the updated version (Line \#475, \#483, highlighted in cyan).

---

> > ### Comment · Reviewer_4sdF · 2024-11-25
> > **Thanks for the response**
> >
> > Thanks for the new results. It would be better if the authors can further compare the most recent papers using new benchmarks, such as:
> > Text Fluoroscopy: Detecting LLM-Generated Text through Intrinsic Features. https://aclanthology.org/2024.emnlp-main.885.pdf
> > Beemo: Benchmark of Expert-edited Machine-generated Outputs.  https://arxiv.org/pdf/2411.04032
> > DLAD: Improving Logits-based Detector without Logits from Black-box LLMs. https://arxiv.org/abs/2406.05232
> > DetectRL: Benchmarking LLM-Generated Text Detection in Real-World Scenarios. https://arxiv.org/pdf/2410.23746

---

> > > ### Author Response · Authors · 2024-11-26
> > >
> > > Thanks for your reply! For the suggested papers, we have conducted the DALD: Improving Logits-based Detector without Logits from Black-box LLM [NeurIPS 24].
> > >
> > > The comparative results are presented below:
> > >
> > > ### Table 1: Comparative AUROC to DALD on RAID data (non-attack) in zero-shot setting
> > > | Detector (token #256)       | Eng$^{mix1}$ | Eng$^{mix2}$ | Eng$^{mix3}$|
> > > |-----------------------------|--------------------|--------------------|--------------------|
> > > | DALD [NeuRIPS'24]          | 0.83              | 0.73              | 0.79              |
> > > | R-Detect[ours]             | **1.00**          | **0.82**          | **0.88**          |
> > >
> > > ### Table 2: Comparative AUROC to DALD on RAID under attack in zero-shot setting
> > > | Detector (token #256)       | Eng$^{mix1}_{att}$ | Eng$^{mix2}_{att}$ | Eng$^{mix3}_{att}$ |
> > > |-----------------------------|-----------------------------------|-----------------------------------|-----------------------------------|
> > > | DALD [NeuRIPS'24]          | 0.71                             | 0.53                             | 0.60                             |
> > > | R-Detect[ours]             | **0.95**                         | **0.57**                         | **0.74**                         |
> > >
> > > We are still working on running more experiments, as is mentioned in "Experiment on the way". However, due to time constraints, we feel very anxious that we might not have enough time to report results on the new benchmarks, especially for results of existing baselines (some of them are running very slowly). But we will try our best.

---

> > > > ### Author Response · Authors · 2024-11-27
> > > >
> > > > Dear Reviewer 4sdF,
> > > >
> > > > The results of using different feature extractors have been updated and you can refer to section Newly added experiments: result summary, design and notations (https://openreview.net/forum?id=z9j7wctoGV&noteId=TISnKQzKS2) for more details.
> > > >
> > > > For your convenience, we have summarized the key findings below:
> > > >
> > > > **Table r1**: Our method achieves the best performance across all baselines on complex MGT detection tasks (based on baselines and tasks you suggested).
> > > >
> > > > **Table r2**: Compared to other baselines, our method demonstrates robustness against adversarial attacks (as per the baseline and tasks you suggested).
> > > >
> > > > **Table r3**: Although our method shows suboptimal performance in non-English detection tasks under a zero-shot setting, detection effectiveness improves significantly with an optimized kernel (e.g., increasing from 0.3 to 0.61 on mix-1, and from 0.35 to 0.43 on mix-2).
> > > >
> > > > **Table r4**: In our default setting from the original submission, the roberta-base-openai-detector was used as the feature extractor. We have now compared it with other available extractors, highlighting the best extractor for each dataset in bold. Notably, with a more powerful extractor such as falcon-rw-1b (while Bino uses falcon-7b), performance can be further enhanced.
> > > >
> > > > If you feel our responses have satisfactorily addressed your concerns, it would be greatly appreciated if you could raise your score to show that the existing concerns have been addressed.
> > > >
> > > > Thank you!
> > > >
> > > > The Authors

---

> > > > > ### Author Response · Authors · 2024-11-29
> > > > >
> > > > > Dear Reviewer 4sdF,
> > > > >
> > > > > Thank you for taking your time to provide us with your valuable comments, which helped improve our paper.
> > > > >
> > > > > We are still waiting for more experimental results with adding---testing new baselines and benchmarks, including two baselines Text Fluoroscopy [EMNLP'24]' and DLAD [NeurRIPS'24]' and two benchmarks Beemo [arXiv'24] and DetectRL`[NeurRIPS'24]'.  However, due to time constraints, we feel very anxious that we might not have enough time to report results on the new benchmarks, especially for results of existing baselines (some of them are running very slowly). We will try to present at least some of them in the rebuttal round.
> > > > >
> > > > > We would like to gently remind you that the discussion period is nearing its conclusion, we hope you have taken the time to consider our responses to your review. If you have any additional questions or concerns, please let us know so we can resolve them before the discussion period concludes. If you feel our responses have satisfactorily addressed your concerns, it would be greatly appreciated if you could raise your score to show that the existing concerns have been addressed.
> > > > >
> > > > > Thank you!
> > > > >
> > > > > The Authors

---

> ### Author Response · Authors · 2024-11-30
> **Results on Beemo benchmark have been updated**
>
> Dear Reviewer 4sdF,
>
> Following your suggestion, we have conducted experiments on the Beemo benchmark, which contains paired HWT and MGT to the same expert question. In addition, the MGT will be rewritten by a different LLM with prompts asking for more human-like texts. Specifically,
>
> > Before we give the results, here is the notation of data:
> >- **mix1**: represents a mixture of MGTs generated by `gemma-7b-it`, `tulu-2-7b`, `gemma-2b-it`, `tulu-2-13b`.
> >- **mix2**: represents a mixture of MGTs generated by `Llama-2-13b-chat-hf`, `Llama-2-70b-chat`, `Mixtral-8x7B-Instruct-v0.1`, `tulu-2-7b`.
> >- **mix3**: represents a mixture of MGTs generated by `tulu-2-13b`, `Mixtral-8x7B-Instruct-v0.1`, `gemma-2b-it`, `Llama-2-13b-chat-hf`, `gemma-7b-it`.
> >
> >- **att#**: `LLM#1|LLM#2` means using `LLM#2` to rewrite the `LLM#1`-generated texts by prompts asking for more human-like texts.
> >  - **att1**: represents a mixture of MGTs generated by:
> >    - `gemma-7b-it|llama-3.1-70b/gpt-4o`
> >    - `tulu-2-7b|gpt-4o`
> >    - `gemma-2b-it|llama-3.1-70b`
> >    - `tulu-2-13b|llama-3.1-70b`.
> >  - **att2**: represents a mixture of MGTs generated by:
> >    - `Llama-2-13b-chat-hf|llama-3.1-70b`
> >    - `Llama-2-70b-chat-hf|llama-3.1-70b`
> >    - `Mixtral-8x7B-Instruct-v0.1|gpt-4o`
> >    - `Llama-2-13b-chat-hf|gpt-4o`
> >    - `tulu-2-7b|gpt-4o`.
> >  - **att3**: represents a mixture of MGTs generated by:
> >    - `tulu-2-13b|gpt-4o`
> >    - `Mixtral-8x7B-Instruct-v0.1|gpt-4o`
> >    - `gemma-2b-it|llama-3.1-70b`
> >    - `Llama-2-13b-chat-hf|gpt-4o`
> >    - `gemma-7b-it|gpt-4o`.
>
> The results are in Table r5.
> ### Table r5: Test on Benchmark Beemo [arXiv24]
>
> | Detector (token #256) | Beemo$^{mix1}$ | Beemo$^{mix2}$| Beemo$^{mix3}$| Beemo$^{mix1}_{att1}$ | Beemo$^{mix2}_{att2}$ | Beemo$^{mix2}_{att2}$ |
> |------------------------|----------------------------------|-----------------------|-----------------------|---------------------------------------------|-------------------------------------|-------------------------------------|
> | MPP [ICLR'24]         | 0.52                             | 0.54                  | 0.56                  | 0.64                                       | 0.63                                | 0.73                                |
> | DetectGPT [ICML'23]   | 0.57                             | 0.33                  | 0.41                  | 0.22                                       | 0.34                                | 0.84                                |
> | Fast-DetectGPT [ICLR'24] | 0.56                          | 0.60                  | 0.49                  | 0.56                                       | 0.55                                | 0.16                                |
> | DALD [NeuRIPS'24]     | 0.47                             | 0.57                  | 0.64                  | 0.63                                       | 0.67                                | **0.93**                            |
> | DNA-GPT [ICLR'24]     | 0.46                             | 0.64                  | 0.50                  | 0.27                                       | 0.38                                | 0.27                                |
> | R-Detect [ours]       | **0.65**                         | **0.81**              | 0.80                  | **0.80**                                   | **1.00**                            | 0.84                                |
>
> The results demonstrate that our method performs the best compared to other baselines on this benchmark. If you feel our responses have satisfactorily addressed your concerns, it would be greatly appreciated if you could raise your score to show that the existing concerns have been addressed.
>
> Thank you!
>
> The Authors

---

> > ### Author Response · Authors · 2024-12-02
> > **Results on DetectRL benchmark have been updated**
> >
> > Dear Reviewer 4sdF,
> >
> > The results of testing on DetectRL `[NeuRIPS]` are attached below as Table r6, as well as in the section Second round added experiments (Part A: benchmark DetectRL[NeuRIPS'24]) where
> >
> > - **human-mix attack** of benchmark DetectRL[NeuRIPS'24] means the MGT is generated by "replacing one-quarter of the sentences in an LLM-generated text with human-written text at random".
> > - **prompt attack** of benchmark DetectRL[NeuRIPS'24]  is "intended to use carefully designed prompts to guide LLMs in generating text that closely mimics human writing style".
> >
> > ### Table r6: Test on Benchmark [NeuRIPS'24]
> > | Detector (token #256)   | non-attack | human-mix attack1 | human-mix attack2 | prompt attack1 | prompt attack2 |
> > |--------------------------|------------|-------------------|-------------------|----------------|----------------|
> > | MPP [ICLR'24]           | 0.69       | 0.80              | 0.78              | 0.64           | 0.62           |
> > | DetectGPT [ICML'23]     | 0.53       | 0.97              | 0.66              | 0.45           | 0.50           |
> > | Fast-DetectGPT [ICLR'24]| 0.88       | 0.97              | 0.73              | 0.67           | 0.90           |
> > | DALD [NeuRIPS'24]       | 0.90       | 0.85              | 0.82              | 0.90           | 0.94           |
> > | DNA-GPT[ICLR'24]        | 0.67       | 0.84              | 0.78              | 0.91           | 0.71           |
> > | R-Detect[ours]          | **1.00**   | **0.98**          | **0.87**          | **1.00**       | **1.00**       |
> >
> > ***The results demonstrate that our method performs the best compared to other baselines on this benchmark.***
> >
> > We have tested all the experiments you have suggested except for threads-of-subtlety `[ACL'24]`  and Text Fluoroscopy `[EMNLP'24]`. The GitHub of threads-of-subtlety `[ACL'24]` is still **empty** (https://github.com/minnesotanlp/threads-of-subtlety) and the authors haven't replied to our request. Text Fluoroscopy `[EMNLP'24]` needs at least **60GB** GPU memory which is far beyond our device capacity i.e., one **RTX™ 4090 24GB**.
> >
> >
> > If you feel our responses have satisfactorily addressed your concerns, it would be greatly appreciated if you could raise your score to show that the existing concerns have been addressed.
> >
> > Thank you!
> >
> > The Authors

---

> > ### Comment · Reviewer_4sdF · 2024-12-02
> > **Thanks for new results**
> >
> > I appreciate the new results. Although I still think the idea lacks sufficient contributions and novelty over MPP-HWT (ICLR'24) and the method needs to collect many training text from existing LLMs which significantly reduces its applicaton in real practice, the authors do provide many experiments. I thus actually netural to it. I would give 5.5 score to this paper and let AC decide.

---

> > > ### Author Response · Authors · 2024-12-02
> > >
> > > Thank you for your comments!
> > >
> > > > Response to your concern: "**The method needs to collect many training texts** from existing LLMs, which significantly reduces its application in real practice."
> > >
> > > We would like to gently remind you that the newly added experiments for baseline comparisons are conducted in a **zero-shot setting**, as noted in the section *Newly added experiments: result summary, design, and notations (Part - Experimental design)*. Specifically, our method uses the previously trained kernel and existing data as references to directly conduct testing.
> > >
> > > Namely, **our method does not need to collect many training texts from existing LLMs.** This is why our approach is highly time-efficient and can be executed with just one RTX™ 4090 24GB GPU.
> > >
> > > ---
> > >
> > > >Response to your concern: "Lacks sufficient contributions and novelty over MPP-HWT (ICLR'24)."
> > >
> > > It is important to clarify that comparing our method to MPP-HWT [ICLR'24] is not entirely accurate. MPP-HWT is a special case of the broader MPP framework, which also includes MPP-MGT. Specifically:
> > >
> > > - **MPP-HWT** uses HWTs as the reference.
> > > - **MPP-MGT** uses MGTs as the reference.
> > >
> > > However, **MPP itself does not predefine whether HWT or MGT is the optimal choice in practice**. The accuracy of MPP-HWT and MPP-MGT can vary significantly. For example, as shown in **Table 3** of our submission, under the HC3 → TQA-ChatGPT column:
> > >
> > > - MPP-HWT outperforms MPP-MGT in terms of the true positive rate (tp$^{MGT}$) (i.e., 1.00 vs. 0.814).
> > > - However, MPP-MGT outperforms MPP-HWT in terms of the false positive rate (tp$^{HWT}$) (i.e., 0.17 vs. 0.88).
> > >
> > > ### Zero-shot testing results on TQA-ChatGPT
> > >
> > > | **Non-parametric Detectors** | **$tp^{\text{MGT}}$**     | **$tp^{\text{HWT}}$**     | **AUROC**       |
> > > |------------------------------|---------------------------|---------------------------|-----------------|
> > > | MPP-HWT                     | $1.00$ (MPP-HWT better than MGT-MGT)        | $0.88$          | $0.77$ |
> > > | MPP-MGT                     | $0.814$        | $0.17$  (MPP-MGT better than MPP-HWT)        | $0.73$ |
> > > | **Our Method (Ours)**        | **$\mathbf{1.00}$** | **$\mathbf{0.00}$** | **$\mathbf{1.00}$** |
> > >
> > >
> > > Our method does not have this drawback. This highlights the significance of our contribution: **relative testing** outperforms the traditional **two-sample testing** approach. Specifically, as noted in Line #105 of our paper: "The MGT detection task is conceptualized as a relative test problem, providing enhanced detection accuracy and flexibility compared to the traditional two-sample test method."
> > >
> > > This contribution has been recognized positively by other reviewers.

---

> > > > ### Author Response · Authors · 2024-12-02
> > > > **Comparions to Text-Fluoroscopy [EMNLP' 24] have been updated**
> > > >
> > > > The authors of Text-Fluoroscopy [EMNLP' 24] have kindly helped us to re-run their code on the tested benchmarks. Due to the random shuffle, we have to list the results in a separate table. Due to the limited time, we only run MPP, our method in this comparison while other baselines are too time costly. The results are shown below, which again shows the excellent performance of our method.
> > > >
> > > > ### Table r7: Comparison to Text-Fluoroscopy [EMNLP' 24] on benchmarks without attack
> > > >
> > > > | Detector (token #256)      | RAID | Beemo | DetectRL |
> > > > |----------------------------|---------------------|---------------------|---------------------|
> > > > | MPP [ICLR' 24]            | 0.74 | 0.52  | 0.69     |
> > > > | Text-Fluoroscopy [EMNLP' 24] | 0.98 | 0.46  | **1.00** |
> > > > | R-Detect [ours]           | **1.00** | **0.65** | **1.00** |
> > > >
> > > > ---
> > > >
> > > > ### Table r8: Comparison to Text-Fluoroscopy [EMNLP' 24] on benchmarks with attack
> > > >
> > > > | Detector (token #256)      | RAID_att | Beemo_att | DetectRL_human_mix | DetectRL_prompt |
> > > > |----------------------------|---------------------|---------------------|---------------------|---------------------|
> > > > | MPP [ICLR' 24]            | 0.64     | 0.63      | 0.80                | 0.62            |
> > > > | Text-Fluoroscopy [EMNLP' 24] | 0.68     | 0.53      | **0.98**           | **1.00**        |
> > > > | R-Detect [ours]           | **0.70** | **1.00**  | **0.98**           | **1.00**        |

---

> ### Author Response · Authors · 2024-12-04
> **Thanks for updating your score to 6!**
>
> Dear Reviewer 4sdF,
>
> We are happy to see that most of your concerns are addressed well! We also noticed that you updated your score to 6. Many thanks for recognizing our contributions to the field.
>
> We will merge all of our discussions, which can definitely help improve our paper further.
>
> Best regards,
>
> Authors of Submission 4390

---

### Official Review · Reviewer_hh24 · 2024-10-31

**Soundness:** 3
**Presentation:** 3
**Contribution:** 2
**Rating:** 6
**Confidence:** 3

**Summary:**

This paper proposes a non-parametric kernel relative test to detect machine generated text (MGTs) by testing whether it is statistically significant that the distribution of a text to be tested is closer to the distribution of human written text (HWTs) than to the MGTs’  distribution. It improves the current two-sample test-based detection methods, which assumes that HWTs must follow the distribution of seen HWT. The authors further develop a kernel optimization algorithm in relative test to select the best kernel that can enhance the testing capability for MGT detection.

**Strengths:**

The proposed relative test can reduce the false positive rate that has been observed in current two sample tests.

It also proposes a novel method to optimize kernels in relative tests for MGT detection, which significantly improving the effectiveness and efficiency of the detection process.

**Weaknesses:**

While this paper presents some innovative ideas, its significance appears limited. There has been some similar studies such as the ICLR'24 paper "Detecting machine-generated texts by multi-population aware optimization for maximum mean discrepancy.".

The proposed method requires the preparation of both the MGT and HWT datasets. In that case, simply comparing the method with Bino in the experiment is insufficient, and may even be unfair because Bino is a zero-shot method. It is essential for the authors to engage in a comprehensive experimental study, comparing their proposed method with various other detection algorithms.

The method leverages Roberta based GPT-2 as feature generator (line 200).  Please explain the reason, and provide a comparison of this choice with other potential alternatives in both text explanation and experimental results.

**Questions:**

The method relies on several assumptions coming from empirical results (line 88, line 280-). The authors are encouraged to provide additional clarifications, both intuitive and theoretical, to further explain those assumptions.

---

> ### Author Response · Authors · 2024-11-24
> **Replies to Weakness 1 and 2**
>
> **We thank you for reviewing our paper and giving valuable comments. We will include rebuttal discussions in the updated version to make the final version more clear to our readers. We hope the answers below solve your concerns. If your concerns were not cleared, please let us know, and we'll try to clarify them further.**
> >Weakness 1: While this paper presents some innovative ideas, its significance appears limited. There has been **some similar studies** such as the ICLR'24 paper *"Detecting machine-generated texts by multi-population aware optimization for maximum mean discrepancy."*
>
> The **unique contribution** of our method compared to MPP is reflected in the following aspects:
> - **Contribution 1:** We are the **first** study that validates **how relative tests are used and preferred over two-sample tests** in MGT detection, while MPP `[ICLR'24]` uses two-sample tests instead.
> - **Contribution 2:** We are the **first** study that provides a solution to **optimize kernels in relative tests**, while MPP `[ICLR'24]` only focuses on kernel optimization in two-sample tests.
>
> The **significance of Contribution 1** is demonstrated in Section 4.2, where we show the **superior performance** of using relative tests **compared to two-sample tests in the same settings**. The comparative results between R-Detect (i.e., detect by relative test) and MPP-HWT/MPP-MGT (i.e., detect by two-sample test using HWT or MGT reference) in **Table 2** show the advancement of our method when HWTs are from **seen** HWT distributions. Similarly, the comparative results between R-Detect and MPP-HWT/MPP-MGT in **Table 3** show the advancement of our method when HWTs are from **unseen** HWT distributions.
>
> The **significance of Contribution 2** is demonstrated by the comparative results in **Table 2 and 3** among R-Detector, R-Detector w/o K* (i.e., using kernels but not optimizing kernels), and R-Detector $k^m$ (i.e., using kernels optimized by median heuristic bandwidth). The results show that our proposed optimization for relative tests is very effective in MGT detection.
>
> >Weakness 2: The proposed method requires the preparation of both the MGT and HWT datasets. In that case, simply comparing the method with Bino in the **experiment is insufficient**, and may even be **unfair because Bino is a zero-shot method**. It is essential for the authors to engage in a comprehensive experimental study, comparing their proposed method with various other detection algorithms.
>
> - It is worth noting that although Bino is zero-shot, it is based on the output of pre-trained LLMs, while our "pre-training" is merely a deep kernel using an NVIDIA RTX graphics card with 16 GB of memory.
>
> - It is also worth noting that we only **trained the kernel once with HC3 and tested it for ALL experiments**. Fairly speaking, the experiments in **Section 4.2.2** (when HWTs are from unseen distributions) and **Section 4.4** (detecting GPT4-rewritten texts) can be considered **zero-shot** testing. Clearly, these are more challenging tasks.
>
> We are conducting more experiments,where we use the **same kernel trained from HC3.** Following Reviewer `[4sdF]`'s comments, the experiments will include:
> 1. **More baselines** such as few-shot detection `[ICLR'24]`, threads-of-subtlety (awaiting code availability) `[ACL'24]`, Detect-GPT `[ICML'23]`, Fast-DetectGPT `[ICLR'24]`, DNA-GPT `[ICLR'24]`, DALD `[NeurIPS'24]`.
> 2. Testing on the benchmark proposed in RAID `[ACL'24]`, which includes data with **adversarial attacks** and **non-English texts.**
> 3. Testing **feature generators other than Roberta-based GPT-2**, such as ChatGPT Detector `[r1]` and Falcon-7B's feature generator (e.g., Bino's).
> `[r1]`: *How close is ChatGPT to human experts? Comparison corpus, evaluation, and detection.* ArXiv 2023.
>
> In the above three experiments, we **directly used the previously trained kernel** from HC3 and **didn't train the kernel** with new data. Additionally, we will conduct
>
> 4. the effectiveness of our method when **training the kernel with RAID** data.
>
> The results will be updated gradually in rebuttal replies and updated to the submission.

---

> > ### Author Response · Authors · 2024-11-24
> > **Replies to Weakness 3 and Question**
> >
> > >Weakness 3: The method leverages **Roberta-based GPT-2** as the feature generator (line 200). Please explain the **reason**, and provide a comparison of this choice with other potential alternatives in both text explanation and experimental results.
> >
> > As MPP `[ICLR'24]` is the most relevant baseline to our method, we directly inherit its feature generator for fair comparisons. As mentioned above, we will test feature generators other than Roberta-based GPT-2 to validate that the superior performance of our method is from the methodology design rather than tricky experimental setups.
> >
> > > Question: The method relies on **several assumptions** coming from empirical results (line 88, line 280-). The authors are encouraged to provide additional clarifications, both intuitive and theoretical, to further explain those assumptions.
> >
> > The assumption in line 88 and line 280 is about the adaptability of kernel optimization from 2ST to relative tests. Theoretical proof of this adaptability is challenging and is not the focus of this paper. Therefore, we gave an intuitive explanation by elaborating the connection between Test Power and TPR/FPR (Line #206, highlighted in cyan).
> > **Test Power *v.s.* TPR *v.s.* FPR.** In hypothesis testing, the test power is defined as the probability of rejecting the null hypothesis when the alternative hypothesis is true. In our method, the null hypothesis is formulated as:
> >
> > \[
> > \mathrm{MMD}(\mathbb{Q}, \mathbb{Z}; k) \leq \mathrm{MMD}(\mathbb{P}, \mathbb{Z}; k),
> > \]
> >
> > which implies that the text to be tested is an MGT. Consequently, the **test power** in our method, given that the ground truth is an MGT, is the **probability that an MGT is correctly identified as an MGT**, corresponding to TPR. Similarly, the **test power** in our method, given that the ground truth is an HWT, is the **probability that an HWT is incorrectly identified as an MGT**, corresponding to FPR.
> >
> > We slightly revised Line #280 to:
> > *"By comparing the empirical test power of our method in Section 3.1: Test Power v.s. TPR v.s. FPR with the 2ST case (Table 1),"* so that readers can make direct connections between Line #206 and #280 (highlighted in cyan).

---

> > > ### Author Response · Authors · 2024-11-27
> > > **Experimental results are updated**
> > >
> > > Dear Reviewer hh24,
> > >
> > > We run more experiments. The performance among methods/data is presented in the tables under section **Newly added experiments: result summary, design and notations** (https://openreview.net/forum?id=z9j7wctoGV&noteId=TISnKQzKS2), using AUROC as the metric. We list the summary here for your convenience.
> > >
> > > ### Summary:
> > > 1. **Table r1**: our method performs the best over all the baselines on complex MGT detection tasks (using baselines and tasks suggested by Reviewer `[4sdF]`).
> > > 2. **Table r2**: our method is robust to adversarial attacks, compared to other baselins (using baseline and tasks suggested by Reviewer `[4sdF]`).
> > > 3. **Table r3**: while our method does not perform well in non-English detection tasks in a zero-shot setting, detection effectiveness improves significantly with a properly optimised kernel (e.g., from 0.3 to 0.61 on mix-1, and 0.35 to 0.43 on mix-2).
> > > 3. **Table r4**: in our default setting in the original submission, roberta-base-openai-detector is the feature extractor. We compare it other available feature extracts and highlight the best extractor for each data in bold. It can be seen that when we use a more powerful extractor, such as falcon-rw-1b (noting that Bino uses falcon-7b), the performance can be further improved.
> > >
> > > ### Experiment on the way:
> > > 1. The TBC values in Table r3 will be gradually replaced.
> > > 3. The same experiments will run with a token size of #512.
> > >
> > > Once all these are done, the experimental results and analysis will be added to the main draft and Appendix.
> > >
> > > If you have any additional questions or concerns, please let us know.

---

> ### Author Response · Authors · 2024-11-28
>
> Dear Reviwer hh24,
>
> Thank you for taking your time to provide us with your valuable comments, which helped improve our paper.
>
> We have replied to your concerns and questions below in section **Replies to Weakness 1 and 2**.
>
> The newly added experiment results are presented in section **Newly added experiments: result summary, design and notations**, and we also validate the analysis summary in section **Experimental results are updated**.
>
> We are still waiting for more experimental results which have been listed in section **Replies to Weakness 1 and 2**.
>
> We would like to gently remind you that the discussion period is nearing its conclusion, we hope you have taken the time to consider our responses to your review. If you have any additional questions or concerns, please let us know so we can resolve them before the discussion period concludes. If you feel our responses have satisfactorily addressed your concerns, it would be greatly appreciated if you could raise your score to show that the existing concerns have been addressed.
>
> Thank you!
>
> The Authors

---

> > ### Author Response · Authors · 2024-12-04
> >
> > Dear Reviewer hh24,
> >
> > We now have completed all experiments. Please refer to the section **Newly added experiments: result summary, design and notations** (the second comment for our paper, from top to bottom).
> >
> > Please see the summary of experiments below (the results are presented in Tables r1–r8 under the section **Newly added experiments: result summary, design, and notations**).
> >
> > - **Table r1**: Our method achieves the best performance on complex MGT detection tasks.
> > - **Table r2**: Our method demonstrates robustness against adversarial attacks.
> > - **Table r3**: While our method initially underperforms in non-English detection tasks in a zero-shot setting, detection effectiveness improves significantly with a properly optimized kernel (e.g., from 0.3 to 0.61 on mix-1, and 0.35 to 0.43 on mix-2).
> > - **Table r4**: Using a powerful extractor, such as falcon-rw-1b (noting that Bino uses falcon-7b) can further enhance performance.
> > - **Table r5**: Demonstrates that our method outperforms other baselines on the **Beemo** [arXiv'24] benchmark.
> > - **Table r6**: Shows that our method achieves superior results on the **DetectRL** [NeurIPS'24] benchmark.
> > - **Tables r7–r8**: Confirms that our method surpasses **Text Fluoroscopy** [EMNLP'24] across the **RAID**, **Beemo**, and **DetectRL** benchmarks, both with and without attacks.
> >
> > We believe these results will make our paper more solid and are further evidence that our paper has good empirical performance in MGT detection (containing comparisons with the newest papers, like papers published in NeurIPS 2024, these papers are actually published after ICLR 2025 submission deadline). We hope these results can help address your major concerns related to our paper. Thanks for your valuable comments again and expect an updated score if your major concerns are addressed by our new results ^^.
> >
> > Best regards,
> >
> > Authors of Submission 4390

---

### Comment · Area_Chair_r7Lt · 2024-11-25

Dear Reviewers,

Thank you for your time and effort in reviewing for ICLR'25.

This is a gentle reminder to read and respond to the authors' rebuttals. Please submit your feedback in time. Thank you very much!

Best regards,

AC

---

### Author Response · Authors · 2024-11-25
**Newly added experiments: result summary, design and notations**

We run more experiments. The performance among methods/data is presented in the following tables, using AUROC as the metric. We list the summary at the beginning, then design, and the detailed results are followed.

### Summary:
1. **Table r1**: our method performs the best on complex MGT detection tasks (using tasks suggested by Reviewer `[4sdF]`).
2. **Table r2**: our method is robust to adversarial attacks (using tasks suggested by Reviewer `[4sdF]`).
3. **Table r3**: while our method does not perform well in non-English detection tasks in a zero-shot setting, detection effectiveness improves significantly with a properly optimised kernel (e.g., from 0.3 to 0.61 on mix-1, and 0.35 to 0.43 on mix-2).
4. **Table r4**: we highlight the best extractor for each data. It can be seen that when we use a powerful extractor, such as falcon-rw-1b (noting that Bino uses falcon-7b), the performance can be further improved.
5. **Table r5**: The results demonstrate that our method performs the best compared to other baselines on the benchmark Beemo `[arXiv24]`.
6. **Table r6**: The results demonstrate that our method performs the best compared to other baselines on the benchmark DetectRL `[NeuRIPS'24]`.
7. **Table r7-8**: The results demonstrate that our method performs the best compared to Text Fluoroscopy `[EMNLP'24]` on the benchmark RAID, Beemo and DetectRL `[NeuRIPS'24]` with and without attack.


### Experiments unavailable:
1. The GitHub of threads-of-subtlety `[ACL'24]` is still **empty** (https://github.com/minnesotanlp/threads-of-subtlety) and the authors haven't replied to our request.

The same experiments will run with a token size of #512 if it is available. All the experimental results and analysis will be added to the main submission and Appendix.

>### Experimental design:
>**Zero-shot setting**: Use the previous trained kernel and reference.
>- **Table r1**: Comparison to more baselines on RAID data (non-attack).
>- **Table r2**: Comparison to more baselines on RAID data under attack.
>- **Table r3 (zero-shot part)**: Test on RAID non-English data with and without attacks.
>- **Table r5**: Test on Beemo benchmark.
>- **Table r6**: Test on DetectRL benchmark.
>- **Table r7-8**: Test and compare to Text Fluoroscopy `[EMNLP'24]` where one of the authors of Text Fluoroscopy `[EMNLP'24]` helped in implementation.
>
> **New kernel setting**: Optimise the kernel with English RAID.
>- **Table r3 (rows of new kernel)**: Test on RAID non-English data with and without attacks with re-trained kernel.
>- **Table r4**: Test with other feature extractors on RAID data with and without attacks with re-trained kernel.
>
> **New reference setting**: Optimise the kernel with English RAID and use unseen RAID data as reference.
>- **Table r3 (rows of new kernel+reference)**: Test on RAID non-English data with and without attacks with re-trained kernel as unseen RAID as reference.
>---
>
>### Notations of datasets:
>RAID provides MGTs generated by different LLMs under different adversarial attacks. We randomly sample MGTs and attacks to consist of the test data, and this test data will be used for testing all detectors. Specifically:
>- **mix1**: Represents a mixture of MGTs generated by Mistral 7B (Chat), GPT-2 XL, MPT-30B, MPT-30B (Chat), and GPT-4.
>- **mix2**: Represents a mixture of MGTs generated by Cohere, Cohere (Chat), GPT-3 (text-davinci-003), and GPT-2 XL.
>- **mix3**: Represents a mixture of MGTs generated by MPT-30B, GPT-3 (text-davinci-003), GPT-2 XL, and GPT-4.
>- **att**: Represents randomly shuffled attacks from Article Deletion, Homoglyph, Number Swap, Paraphrase, Synonym Swap, Misspelling, Whitespace Addition, Upper-Lower Swap, Zero-Width Space, Insert Paragraphs, Alternative Spelling. Namely, a misspelled sentence can be followed by a synonym swapped sentence.
>- **N/Eng**: Represents using random samples of non-English MGTs (Code, Czech, German) in RAID.

---

> ### Author Response · Authors · 2024-11-25
> **Experiment results and brief analysis (updating)**
>
> ### Table r1: Comparative AUROC to more baselines on RAID data (non-attack) in zero-shot setting
>
> | Detector (token #256)       | Eng$^{mix1}$ | Eng$^{mix2}$ | Eng$^{mix3}$|
> |-----------------------------|--------------------|--------------------|--------------------|
> | MPP [ICLR'24]              | 0.74              | 0.70              | 0.85              |
> | DetectGPT [ICML'23]        | 0.64              | 0.66              | 0.50              |
> | Fast-DetectGPT [ICLR'24]   | 0.84              | 0.81              | 0.63              |
> | DALD [NeuRIPS'24]          | 0.83              | 0.73              | 0.79              |
> | DNA-GPT[ICLR'24]           | 0.81              | 0.36              | 0.32              |
> | R-Detect[ours]             | **1.00**          | **0.82**          | **0.88**          |
>
> ---
>
> ### Table r2: Comparative AUROC to more baselines on RAID under attack in zero-shot setting
>
> | Detector (token #256)       | Eng$^{mix1}_{att}$ | Eng$^{mix2}_{att}$ | Eng$^{mix3}_{att}$ |
> |-----------------------------|-----------------------------------|-----------------------------------|-----------------------------------|
> | MPP [ICLR'24]              | 0.71                             | 0.56                             | 0.70                             |
> | DetectGPT [ICML'23]        | 0.59                             | 0.51                             | 0.46                             |
> | Fast-DetectGPT [ICLR'24]   | 0.44                             | 0.41                             | 0.39                             |
> | DALD [NeuRIPS'24]          | 0.71                             | 0.53                             | 0.60                             |
> | DNA-GPT[ICLR'24]           | 0.71                             | 0.21                             | 0.39                             |
> | R-Detect[ours]             | **0.95**                         | **0.57**                         | **0.74**                         |
>
> ---
>
> ### Table r3: Test on non-English RAID data (with and without attack) in different settings
>
> | Detector (token #256)       | Setting            | N/Eng$^{mix1}$ | N/Eng$^{mix1}_{att}$ | N/Eng$^{mix2}$ | N/Eng$^{mix2}_{att}$ |
> |-----------------------------|--------------------|----------------------|-----------------------------------|----------------------|-----------------------------------|
> | MPP                        | Zero-shot          | **0.47**            | 0.43                             | **0.45**            | 0.39                             |
> | R-Detect                   | Zero-shot          | 0.30                | **0.56**                         | 0.35                | **0.45**                         |
> | MPP                        | New kernel         | 0.56                | 0.54                             | **0.47**            | **0.43**                              |
> | R-Detect                   | New kernel         | **0.61**            | **0.78**                         | 0.43                | 0.36                              |
> | MPP                        | Kernel + Reference | 0.61               | 0.43                             | 0.47                 |**0.57**                              |
> | R-Detect                   | Kernel + Reference |**0.95**               | **0.80**                         |**0.77**                 | 0.54                              |
>
> ---

---

> ### Author Response · Authors · 2024-11-27
> **Experiment results  (Part B)**
>
> ### Table r4: Test with different feature extractors
>
> | Feature Extractor (token #256)          |       | Eng$^{mix1}$ | Eng$^{mix1}_{att}$ | Eng$^{mix2}$ | Eng$^{mix2}_{att}$ |
> |-----------------------------------------|-------|-----------------------|-------------------------------------|-----------------------|-------------------------------------|
> | **roberta-base-openai-detector**        | MPP   | 0.64                  | 0.60                                | 0.58                  | 0.60                                |
> |                                         | R-Detect[ours] | **1.00**          | **0.94**                            | 0.94                  | 0.79                                |
> | **falcon-rw-1b**                        | MPP   | **0.80**              | 0.57                                | **0.77**              | **0.71**                            |
> |                                         | R-Detect[ours] | 0.98              | **0.81**                            | **0.99**              | **0.86**                            |
> | **roberta-base**                        | MPP   | 0.71                  | 0.60                                | 0.63                  | 0.41                                |
> |                                         | R-Detect[ours] | **1.00**          | 0.77                                | 0.97                  | 0.69                                |
> | **chatgpt-detector-roberta**            | MPP   | 0.58                  | 0.59                                | 0.62                  | 0.56                                |
> |                                         | R-Detect[ours] | **1.00**          | 0.86                                | 0.90                  | 0.79                                |

---

> ### Author Response · Authors · 2024-11-30
> **Second round added experiments (Part A: Beemo)**
>
> Following Reviewer `[4sdF]`'s reply, two baselines and two benchmarks are suggested where the result of DALD `[NeurIPS'24]` has been attached in the first round added experiments. In this section, we give the experimental results on one of the benchmark Beemo `[arXiv'24]` which contains paired HWT and MGT to the same expert question. In addition, the MGT will be rewritten by a different LLM with prompts asking for more human-like texts. Specifically,
>
> Notes of datasets:
> - **mix1**: represents a mixture of MGTs generated by `gemma-7b-it`, `tulu-2-7b`, `gemma-2b-it`, `tulu-2-13b`.
> - **mix2**: represents a mixture of MGTs generated by `Llama-2-13b-chat-hf`, `Llama-2-70b-chat`, `Mixtral-8x7B-Instruct-v0.1`, `tulu-2-7b`.
> - **mix3**: represents a mixture of MGTs generated by `tulu-2-13b`, `Mixtral-8x7B-Instruct-v0.1`, `gemma-2b-it`, `Llama-2-13b-chat-hf`, `gemma-7b-it`.
>
> - **att#**: `LLM#1|LLM#2` means using `LLM#2` to rewrite the `LLM#1`-generated texts by prompts asking for more human-like texts.
>   - **att1**: represents a mixture of MGTs generated by:
>     - `gemma-7b-it|llama-3.1-70b/gpt-4o`
>     - `tulu-2-7b|gpt-4o`
>     - `gemma-2b-it|llama-3.1-70b`
>     - `tulu-2-13b|llama-3.1-70b`.
>   - **att2**: represents a mixture of MGTs generated by:
>     - `Llama-2-13b-chat-hf|llama-3.1-70b`
>     - `Llama-2-70b-chat-hf|llama-3.1-70b`
>     - `Mixtral-8x7B-Instruct-v0.1|gpt-4o`
>     - `Llama-2-13b-chat-hf|gpt-4o`
>     - `tulu-2-7b|gpt-4o`.
>   - **att3**: represents a mixture of MGTs generated by:
>     - `tulu-2-13b|gpt-4o`
>     - `Mixtral-8x7B-Instruct-v0.1|gpt-4o`
>     - `gemma-2b-it|llama-3.1-70b`
>     - `Llama-2-13b-chat-hf|gpt-4o`
>     - `gemma-7b-it|gpt-4o`.
>
> ### Table r5: Test on Benchmark Beemo [arXiv24]
>
> | Detector (token #256) | Beemo$^{mix1}$ | Beemo$^{mix2}$| Beemo$^{mix3}$| Beemo$^{mix1}_{att1}$ | Beemo$^{mix2}_{att2}$ | Beemo$^{mix2}_{att2}$ |
> |------------------------|----------------------------------|-----------------------|-----------------------|---------------------------------------------|-------------------------------------|-------------------------------------|
> | MPP [ICLR'24]         | 0.52                             | 0.54                  | 0.56                  | 0.64                                       | 0.63                                | 0.73                                |
> | DetectGPT [ICML'23]   | 0.57                             | 0.33                  | 0.41                  | 0.22                                       | 0.34                                | 0.84                                |
> | Fast-DetectGPT [ICLR'24] | 0.56                          | 0.60                  | 0.49                  | 0.56                                       | 0.55                                | 0.16                                |
> | DALD [NeuRIPS'24]     | 0.47                             | 0.57                  | 0.64                  | 0.63                                       | 0.67                                | **0.93**                            |
> | DNA-GPT [ICLR'24]     | 0.46                             | 0.64                  | 0.50                  | 0.27                                       | 0.38                                | 0.27                                |
> | R-Detect [ours]       | **0.65**                         | **0.81**              | 0.80                  | **0.80**                                   | **1.00**                            | 0.84                                |

---

> ### Author Response · Authors · 2024-12-01
> **Second round added experiments (Part B: benchmark DetectRL[NeuRIPS'24])**
>
> - **human-mix attack** of benchmark DetectRL[NeuRIPS'24] means the MGT is generated by "replacing one-quarter of the sentences in an LLM-generated text with human-written text at random".
> - **prompt attack** of benchmark DetectRL[NeuRIPS'24]  is "intended to use carefully designed prompts to guide LLMs in generating text that closely mimics human writing style".
>
> ### Table r6: Test on Benchmark [NeuRIPS'24]
> | Detector (token #256)   | non-attack | human-mix attack1 | human-mix attack2 | prompt attack1 | prompt attack2 |
> |--------------------------|------------|-------------------|-------------------|----------------|----------------|
> | MPP [ICLR'24]           | 0.69       | 0.80              | 0.78              | 0.64           | 0.62           |
> | DetectGPT [ICML'23]     | 0.53       | 0.97              | 0.66              | 0.45           | 0.50           |
> | Fast-DetectGPT [ICLR'24]| 0.88       | 0.97              | 0.73              | 0.67           | 0.90           |
> | DALD [NeuRIPS'24]       | 0.90       | 0.85              | 0.82              | 0.90           | 0.94           |
> | DNA-GPT[ICLR'24]        | 0.67       | 0.84              | 0.78              | 0.91           | 0.71           |
> | R-Detect[ours]          | **1.00**   | **0.98**          | **0.87**          | **1.00**       | **1.00**       |

---

> > ### Author Response · Authors · 2024-12-02
> > **Second round added experiments (Part C: comparions to Text-Fluoroscopy [EMNLP' 24])**
> >
> > The authors of Text-Fluoroscopy [EMNLP' 24] have kindly helped us to re-run their code on the tested benchmarks. Due to the random shuffle, we have to list the results in a separate table. Due to the limited time, we only run MPP, our method in this comparison while other baselines are too time costly. The results are shown below, which again shows the excellent performance of our method.
> >
> > ### Table r7: Comparison to Text-Fluoroscopy [EMNLP' 24] on benchmarks without attack
> >
> > | Detector (token #256)      | RAID | Beemo | DetectRL |
> > |----------------------------|---------------------|---------------------|---------------------|
> > | MPP [ICLR' 24]            | 0.74 | 0.52  | 0.69     |
> > | Text-Fluoroscopy [EMNLP' 24] | 0.98 | 0.46  | **1.00** |
> > | R-Detect [ours]           | **1.00** | **0.65** | **1.00** |
> >
> > ---
> >
> > ### Table r8: Comparison to Text-Fluoroscopy [EMNLP' 24] on benchmarks with attack
> >
> > | Detector (token #256)      | RAID_att | Beemo_att | DetectRL_human_mix | DetectRL_prompt |
> > |----------------------------|---------------------|---------------------|---------------------|---------------------|
> > | MPP [ICLR' 24]            | 0.64     | 0.63      | 0.80                | 0.62            |
> > | Text-Fluoroscopy [EMNLP' 24] | 0.68     | 0.53      | **0.98**           | **1.00**        |
> > | R-Detect [ours]           | **0.70** | **1.00**  | **0.98**           | **1.00**        |

---

### Meta-Review · Area_Chair_r7Lt · 2024-12-19

**Metareview:**

This paper explores the challenge of detecting machine-generated texts. It argues that existing methods fail to identify human-written texts when there is a shift in distribution. The paper proposes using a non-parametric kernel relative test to determine whether the distribution of a given text is statistically closer to that of human-written texts than to machine-generated texts. To improve detection performance, a kernel optimization algorithm is introduced to select the most effective kernel for this task. Experimental results demonstrate the effectiveness of the proposed approach. The paper presents a reasonable motivation, addressing a problem with significant practical value. The proposed method is both simple and effective, with clear logic and a well-explained algorithm. It is built on a solid theoretical foundation, with logically sound arguments and high interpretability. Furthermore, the experimental results demonstrate the algorithm’s strong performance. Therefore, I believe this paper should be accepted.

**Additional Comments On Reviewer Discussion:**

During the discussion phase, two reviewers believed that the paper lacked sufficient experiments and had flaws in its writing logic and expression. However, the authors responsibly addressed the reviewers' comments and resolved their concerns through additional experiments. Ultimately, all reviewers provided positive feedback.

---

### Decision · Program_Chairs · 2025-01-22

Accept (Poster)